# *Lactobacillus paragasseri* LPG-9 reduces placental inflammation in intrahepatic cholestasis of pregnancy by regulating TGR5 in mice
Wanwen Huang[1], Jiechang Zhang[2], Jiamin Shan[2], Wei Shen[3], Pingping Du[4], Jiaxin Liu[2], Xiaotong Guo[2], Zhenhui Chen[2], Weisen Zeng [5]✉, Qiongxi Lin [2]✉ & Hongying Fan [2]✉

Intrahepatic cholestasis of pregnancy (ICP), a liver disorder associated with adverse fetal outcomes, is characterized by elevated bile acid levels and placental inflammation by the TGR5. However, the interplay among the gut microbiome, bile acid metabolism, and ICP-associated placental inflammation remains unexplored. We aimed to investigate the role of the gut microbiota in regulating bile acid metabolism and placental inflammation, and to identify potential probiotic-based therapies for ICP in C57BL/6 mice. Immunohistochemical analysis of human placentas revealed significantly higher inflammation and decreased TGR5 expression in ICP compared with controls. In vivo and in vitro assays confirmed the anti-inflammatory effects of TGR5 activation. Using 16S rRNA sequencing and metabolomics, ICP mice exhibited a distinct gut microbiota composition and reduced abundance of bile salt hydrolase (BSH)-producing bacteria (e.g., *Lactobacillus*), accompanied by a significant decrease in the proportion of secondary bile acids. Transplanting fecal microbiota from ICP donors into healthy mice reproduced the disease phenotype of ICP, confirming the pathogenic role of gut microbiota dysbiosis. Supplementation with BSH-enriched *Lactobacillus paragasseri* LPG-9 remodeled the bile acid profile, thereby activating placental TGR5 to inhibit TLR4-NF-κB signaling and promoting hepatic bile acid excretion via BSEP.

Intrahepatic cholestasis of pregnancy (ICP) is a liver disease associated with severe fetal complications[1]. Elevated maternal serum bile acids are a hallmark of ICP and are strongly correlated with an increased risk of adverse fetal outcomes[2]. Additionally, ICP is characterized by an abnormal bile acid profile dominated by primary bile acids[3,4]. However, the molecular relationship between bile acids and fetal outcomes remains poorly understood.

Elevated serum bile acids trigger placental inflammation by activating the nuclear factor-kappa B (NF-κB) pathway through the Takeda G-protein-coupled receptor 5 (TGR5) in trophoblasts[5,6]. TGR5, a plasma membrane-bound bile acid receptor, is a protective regulatory target in metabolic and inflammatory diseases[7,8]. This receptor is highly expressed in the human term placenta and localized to fetal macrophages. Notably,

TGR5 expression is significantly downregulated in the placentas of patients with ICP and in pregnant cholestatic rats[9]. TGR5 activation is primarily driven by the secondary bile acids lithocholic acid (LCA) and deoxycholic acid (DCA), which are markedly reduced in the serum of patients with ICP.

The synthesis of secondary bile acids is catalyzed by bile salt hydrolases (BSH) and 7α-dehydroxylase, enzymes produced by gut bacteria. Animal experiments and population studies suggest that the gut microbiome plays a crucial role in cholestatic liver disease severity. The gut microbiota composition of patients with ICP differs from that of healthy pregnant women, and the abundance of specific taxa correlates with serum bile acid concentrations and disease severity[10,11]. Moreover, studies in germ-free mice[12] and probiotic interventions[13] have shown that microbiome modulation can

[1]Experimental Teaching Center of Preventive Medicine, Guangdong Provincial Key Laboratory of Tropical Disease Research, School of Public Health, Southern Medical University, Guangzhou, China. [2]Department of Microbiology, Guangdong Provincial Key Laboratory of Tropical Disease Research, School of Public Health, Southern Medical University, Guangzhou, China. [3]Department of Neonatology, Nanfang Hospital, Southern Medical University, Guangzhou, China. [4]Department of Obstetrics and Gynaecology, Nanfang Hospital, Southern Medical University, Guangzhou, China. [5]Department of Cell Biology, School of Basic Medicine Sciences, Southern Medical University, Guangzhou, China. ✉e-mail: zengws@smu.edu.cn; lqx123@smu.edu.cn; gzfhy@smu.edu.cn

regulate bile acid metabolism and reduce hepatic bile acid accumulation. Therefore, we hypothesized that the gut microbiota represents a potential therapeutic target for ICP.

By investigating the relationship among the gut microbiome, bile acid metabolism, and ICP-associated placental inflammation, we aimed to identify candidate probiotics and explore their mechanisms in mitigating placental inflammation through TGR5 regulation. Our study provides mechanistic insights into ICP pathology and supports the development of microbiome-based therapies to improve fetal outcomes.

## Results

### Abnormal expression of the TGR5-TLR4-NF-κB pathway in the placentas of humans and animals with ICP

Hematoxylin-eosin (HE) and myeloperoxidase (MPO) staining revealed increased neutrophil infiltration in the placental tissues of five patients with ICP compared with that in five matched controls (Supplementary Fig. 1a). Western blot analysis demonstrated significantly elevated levels of pro-inflammatory cytokines (TNF-α, IL-6, and IL-1β) in ICP placentas (Fig. 1b and Supplementary Fig. 1b). Immunohistochemistry of the human placenta confirmed significantly higher TLR4 and NF-κB expression in patients with ICP than in controls, whereas TGR5 expression was reduced in ICP (Fig. 1a and Supplementary Fig. 2a). These results suggest that increased placental inflammation in patients with ICP is associated with TGR5 inhibition.

To explore the underlying mechanism, we established a mouse model of ICP with varying serum bile acid levels (Fig. 1c). Total bile acids (TBA) were significantly higher in ICP mice (Fig. 1d), while aspartate aminotransferase (AST) and alanine aminotransferase (ALT) showed no significant differences. Increased TBA levels were associated with reduced offspring survival and birth weight in the ICP group (Fig. 1e). Histological analysis of liver tissue revealed features of cholestasis, including hepatocyte ballooning, inflammatory cell infiltration, and bile plugs, which were particularly pronounced in the 1.0% CA group (Supplementary Fig. 1d). These findings confirm the successful establishment of the model. HE staining revealed placental pathological changes in the ICP group, including evident vascular congestion, increased neutrophil infiltration, and interstitial edema, which worsened with higher TBA levels (Fig. 1f). MPO staining further confirmed significantly increased neutrophil infiltration in the ICP placentas (Supplementary Fig. 2b), a hallmark of placental inflammation[14]. Consistent with human findings, placental TLR4 and NF-κB expression were significantly elevated in ICP mice, while TGR5 and IκB-α expressions were downregulated (Fig. 1g). mRNA levels of *IL-1β*, *IL-6*, and *TNF-α* increased in parallel with rising serum bile acid levels (Fig. 1h).

To determine whether TGR5 induces a placental inflammatory response, we treated the 0.5% cholic acid (CA)-induced ICP model mice with the TGR5 receptor agonist (CCDC). The agonist inhibited placental inflammation (Supplementary Fig. 3) and improved the survival rates and weight of the offspring (Fig. 1i). In addition, TLR4 and NF-κB proteins were downregulated when TGR5 was overexpressed following treatment with the TGR5 receptor agonist (CCDC) in *J774.2* cells (Fig. 1j). To further delineate the interplay between TGR5 and TLR4, we performed co-immunoprecipitation (Co-IP). The interaction between TGR5 and TLR4 proteins was confirmed in endogenous conditions (within *J774.2* macrophages; Fig. 1k) and in an overexpression system involving *HEK293T* cells (Supplementary Fig. 1c). These findings indicate that TGR5 activation suppresses inflammatory responses by downregulating the TLR4-NF-κB signaling pathway in ICP.

### Bile acid metabolism and gut microbiota disturbance in ICP mice

As TGR5 is primarily regulated by bile acids, we performed targeted metabolomics using high-performance liquid chromatography-mass spectrometry (HPLC-MS) to assess bile acid composition in the serum of control and ICP mice. Principal component analysis (PCA) revealed a significantly altered bile acid profile in ICP mice compared with controls (Fig. 2a). The proportion of primary bile acids in the ICP group was significantly increased, whereas that of secondary bile acids was decreased

(Fig. 2b, c). Furthermore, the proportions of glycodeoxycholic acid (GDCA), β-cholic acid (βCA), tauro-ω-muricholic acid (T-ω-MCA), and hyodeoxycholic acid (HDCA) in the serum were significantly decreased in the ICP group (Fig. 2d).

The gut microbiota is associated with secondary bile acid metabolism and cholestasis-related diseases[15]. Therefore, we examined changes in the gut microbiota of the two groups. Principal coordinate analysis (PCoA) showed that overall microbial compositions in the feces differed between the ICP and control groups (Fig. 2e). At the genus level, the abundance of *Lactobacillus* was significantly decreased in the ICP group compared to the control group (Fig. 2f). Linear discriminant analysis revealed that the control group was enriched in *Lactobacillus*, *Oscillospirales*, *Eggerthellaceae*, and *Clostridia*, whereas the ICP group was enriched in *Klebsiella* and *Enterobacter* (Fig. 2h). Furthermore, Tax4Fun predictions indicated reduced bile secretion and bile acid biosynthesis in the ICP group (Fig. 2g). We also examined the expression of the key enzymes catalyzing bile acid metabolism. Notably, *bsh* expression was significantly downregulated in the ICP group (Fig. 2i), while *Bai* expression showed no significant changes. Moreover, *bsh* expression correlated with *Lactobacillus* abundance, suggesting that deficiency of BSH-related *Lactobacillus* contributes to ICP development (Fig. 2j).

### Gut microbiota contributes to the progression of ICP

Fecal microbiota transplantation (FMT) was performed to further investigate the role of the gut microbiota in ICP progression. Recipient mice were orally inoculated with the fecal contents from patients with ICP (Fig. 3a). After FMT, PCoA revealed distinct intestinal microbiome compositions between the two groups (Fig. 3b). Consistent with ICP results, the abundance of *Lactobacillus* (Fig. 3c) and *bsh* expression (Fig. 3e) were significantly reduced in FMT mice. Similarly, Tax4Fun analysis predicted a marked reduction in bile acid synthesis and bile secretion capacity after FMT (Fig. 3d).

In addition, serum TBA levels were significantly increased and offspring survival rates markedly decreased after FMT (Fig. 3f, g). These trends negatively correlated with the relative abundance of *Lactobacillus* (Fig. 3j). Histological analysis revealed increased hepatic inflammatory cells (by HE staining, Supplementary Fig. 4a), as well as increased vascular congestion (by HE staining, Fig. 3h) and increased neutrophil infiltration (by MPO staining, Supplementary Fig. 4b) in the placental tissues following FMT. Consistent with the ICP model, TLR4 and NF-κB expression were elevated in the placenta of FMT mice, while TGR5 and IκB-α were downregulated (Fig. 3i). These findings suggest that the gut microbiota influences bile acid metabolism and contributes to dysregulation of the TGR5-TLR4-NF-κB pathway in patients with ICP.

### LPG-9 improves ICP-mediated adverse offspring outcomes by activating TGR5

Given the decreases in *Lactobacillus* and *bsh* levels in ICP, we hypothesized that BSH-enriched *Lactobacillus* could improve placental inflammation. Among strains isolated from healthy individuals, *Lacticaseibacillus rhamnosus* LRX01 and *Lactobacillus paragasseri* LPG-9 exhibited high BSH activity (Fig. 4a, b). LPG-9 had the highest BSH activity and efficiently reduced serum TBA levels in the cholestatic mouse model (Fig. 4c). Whole-genome sequencing revealed that LPG-9 carried abundant BSH-related genes and lacked pathogenic virulence factors (Fig. 4d), suggesting LPG-9 is both effective and safe for ICP treatment. Subsequent treatment of ICP mice with LPG-9 significantly reduced serum TBA levels and improved offspring survival and birth weight (Fig. 4e, f). HE staining showed improvements in hepatocyte ballooning and inflammatory infiltration after LPG-9 intervention (Supplementary Fig. 4c). The intervention caused increased expression of TGR5 and IκB-α in the placenta, while TLR4 and NF-κB were decreased (Fig. 4g). Placental neutrophil infiltration and inflammatory factors expression were also significantly reduced in the LPG-9 group (Fig. 4h, j and Supplementary Fig. 4d).

We next examined whether LPG-9 modulated serum bile acids composition to enhance TGR5 expression. PCA of serum bile acids revealed distinct profiles among the three groups, with the LPG-9 group resembling

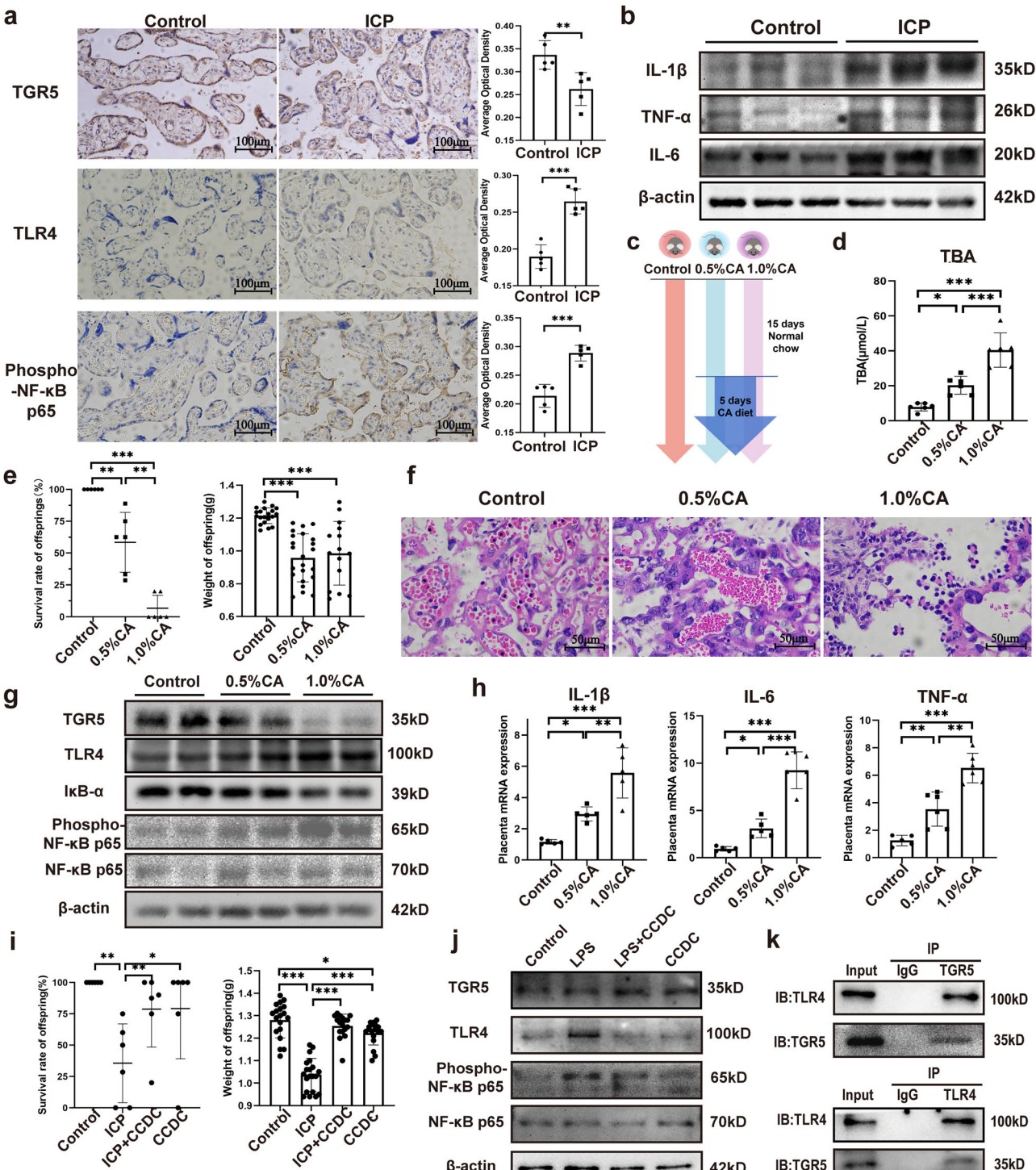

**Fig. 1 | Abnormal expression of the TGR5 pathway in the placenta of humans and mice with ICP. a** Immunohistochemical analysis of TGR5, TLR4 and phospho-NF-κB p65 in the human placental tissue, 200×. Scale bar = 100 μm (*n* = 5 biologically independent samples; unpaired two-tailed *t*-test); **b** Western blotting of pro-inflammatory cytokines in the human placental tissue; **c** Schematic representation of cholic acid (CA)-induced model of ICP; **d** Total serum bile acids (TBA) analysis in the ICP model (*n* = 6 biologically independent samples; one-way ANOVA); **e** Survival and weight of offspring in the ICP model (*n* = 6 biologically independent samples; one-way ANOVA); **f** Hematoxylin-eosin-stained sections of mouse placenta, 400×. Scale bar = 50 μm; **g** Western blotting of placental TGR5, TLR4, and NF-κB; **h** Quantitative polymerase chain reaction (qPCR) data showing the gene expression levels of placental pro-inflammatory cytokines (*n* = 5–6 biologically independent samples; one-way ANOVA); **i** Survival rates and weight of offspring in the ICP model mice treated with a TGR5 receptor agonist (*n* = 6 biologically independent samples; one-way ANOVA); **j** Western blotting of TLR4 and NF-κB in *J774.2* cells treated with a TGR5 receptor agonist; **k** Co-immunoprecipitation of TGR5 and TLR4 protein in *J774.2* cells. Error bars represent standard deviation. Significance levels: *\*p* < 0.05, *\*\*p* < 0.01, *\*\*\*p* < 0.001.

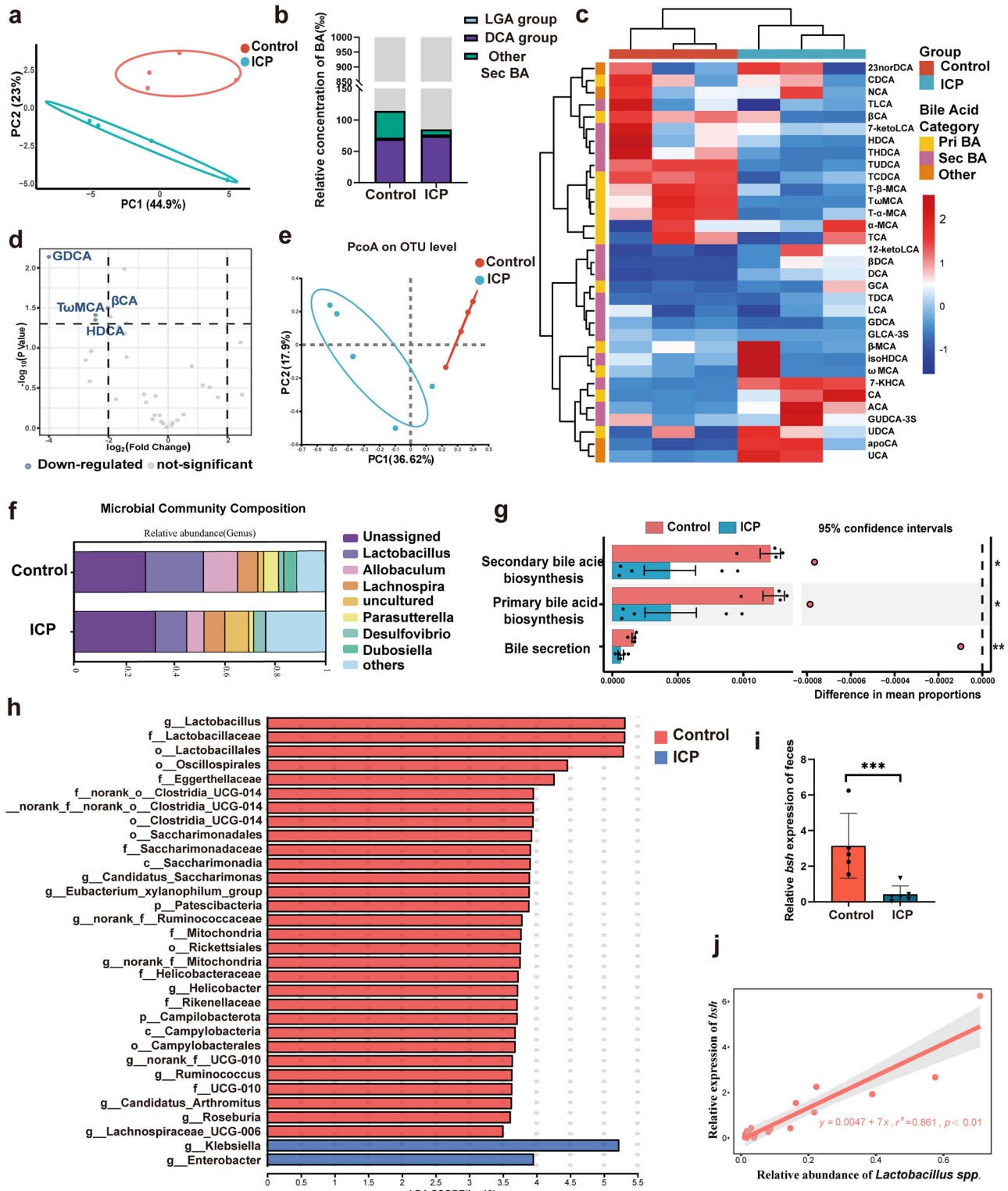

**Fig. 2 | Bile acid metabolism and gut microbiota disturbance in ICP mice.**
**a** Principal component analysis diagram of bile acid in the serum; **b** Stacked bar graph of secondary bile in the serum; **c** Cluster analysis heat map of the serum bile acid. The color scale represents normalized abundance (row Z-score), with blue indicating low abundance and red indicating high abundance; **d** Volcano plot of serum bile acid (ICP vs. Control); **e** Principal coordinate analysis diagram of gut microbiome; **f** Bar diagram of the microbial composition; **g** Results of the KEGG pathway predictions based on Tax4Fun. Error bars represent standard error; **h** Dominant bacteria were significantly different between the control and the ICP groups (LEfSe analysis); **i** bsh expression in feces (n = 5–6 biologically independent samples; unpaired two-tailed t-test). Error bars represent standard deviation; **j** Correlation analysis of bsh expression and the abundance of Lactobacillus. Significance levels: *p < 0.05, **p < 0.01, ***p < 0.001.

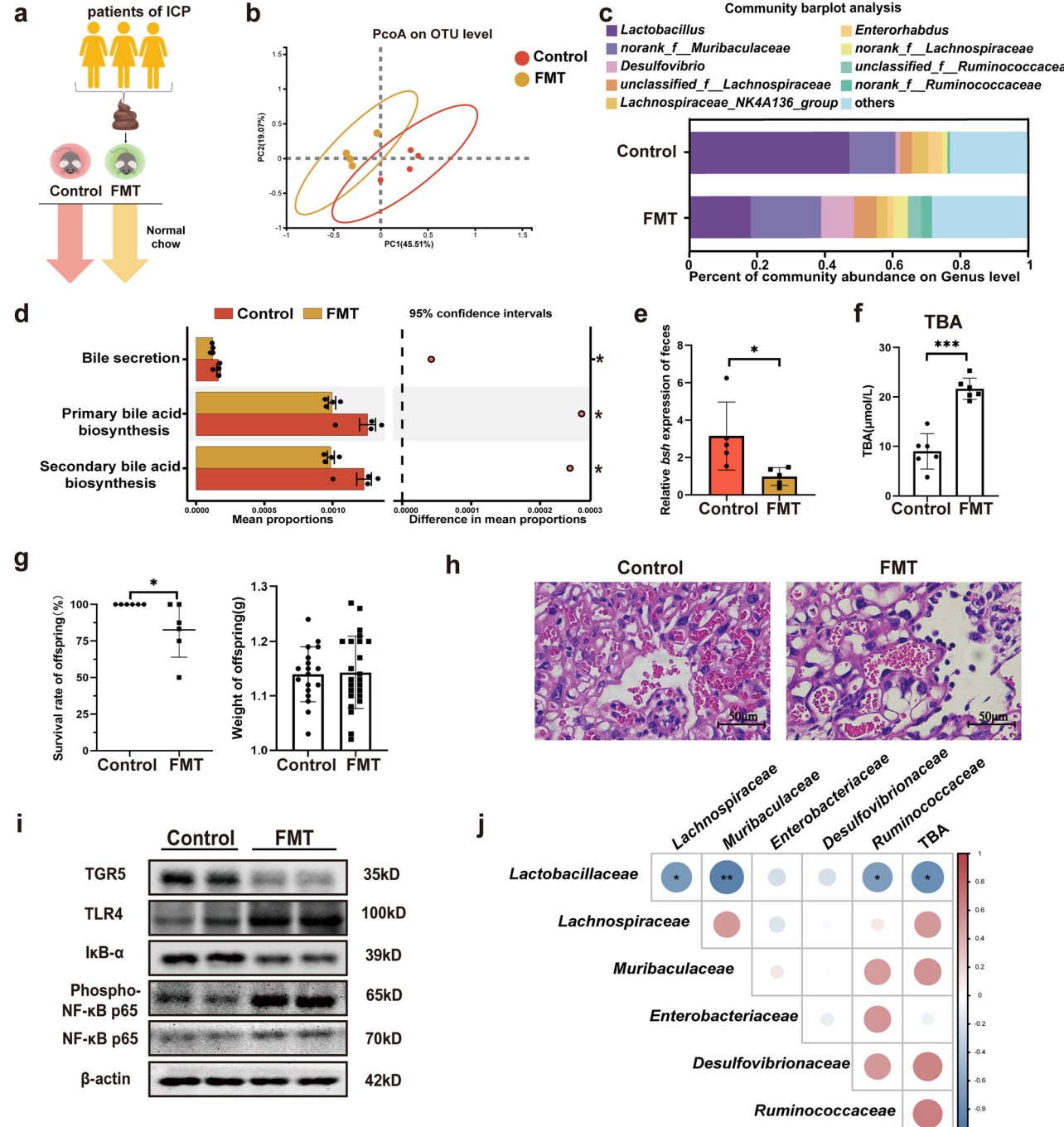

**Fig. 3 | Gut microbiota contributes to the progression of ICP. a** Schematic representation of the fecal microbiota transplantation (FMT) intervention; **b** Principal coordinate analysis diagram of the gut microbiome; **c** Bar diagram of the fecal microbial composition; **d** Results of the KEGG pathway predictions based on Tax4Fun. Error bars represent standard error; **e** *bsh* expression in feces (*n* = 5–6 biologically independent samples; unpaired two-tailed *t*-test). Error bars represent standard deviation; **f** Serum TBA levels were increased after FMT (*n* = 6 biologically independent samples; unpaired two-tailed *t*-test). Error bars represent standard deviation; **g** The survival rates and weight of offspring (*n* = 6 biologically independent samples; unpaired two-tailed *t*-test). Error bars represent standard deviation; **h** Hematoxylin-eosin-stained sections of the placenta after FMT, 400×. Scale bar = 50 μm; **i** Western blotting of TGR5, TLR4, and NF-κB in the FMT mice placenta; **j** Correlation analysis of the most dominant microbial family and serum TBA. The color scale represents Spearman's correlation coefficient, with red indicating positive correlation and blue indicating negative correlation. Significance levels: \**p* < 0.05, \*\**p* < 0.01, \*\*\**p* < 0.001.

the control group (Fig. 4i). Serum proportions of glycocholic acid (GCA), taurodeoxycholic acid (TDCA), GDCA, DCA, 12-ketoLCA, and LCA were significantly increased, whereas tauro-β-muricholic acid (T-β-MCA) decreased (Fig. 4k). Overall, the proportion of secondary bile acids in the serum was significantly increased after LPG-9 treatment (Fig.4l). These results indicate that LPG-9 restores placental TGR5 expression in ICP by reshaping serum bile acid composition.

## LPG-9 restores the gut microbiome profile in ICP and promotes bile acid secretion

PCoA revealed distinct microbial profiles between the LPG-9 and ICP groups, with LPG-9-treated mice resembling controls (Fig. 5a). The reduced abundance of *Lactobacillus* and elevated *Klebsiella* levels were reversed by LPG-9 treatment, indicating that alleviation of the gut microbiota dysbiosis in ICP (Fig. 5b). Tax4Fun predicted increased bile acid secretion and biosynthesis

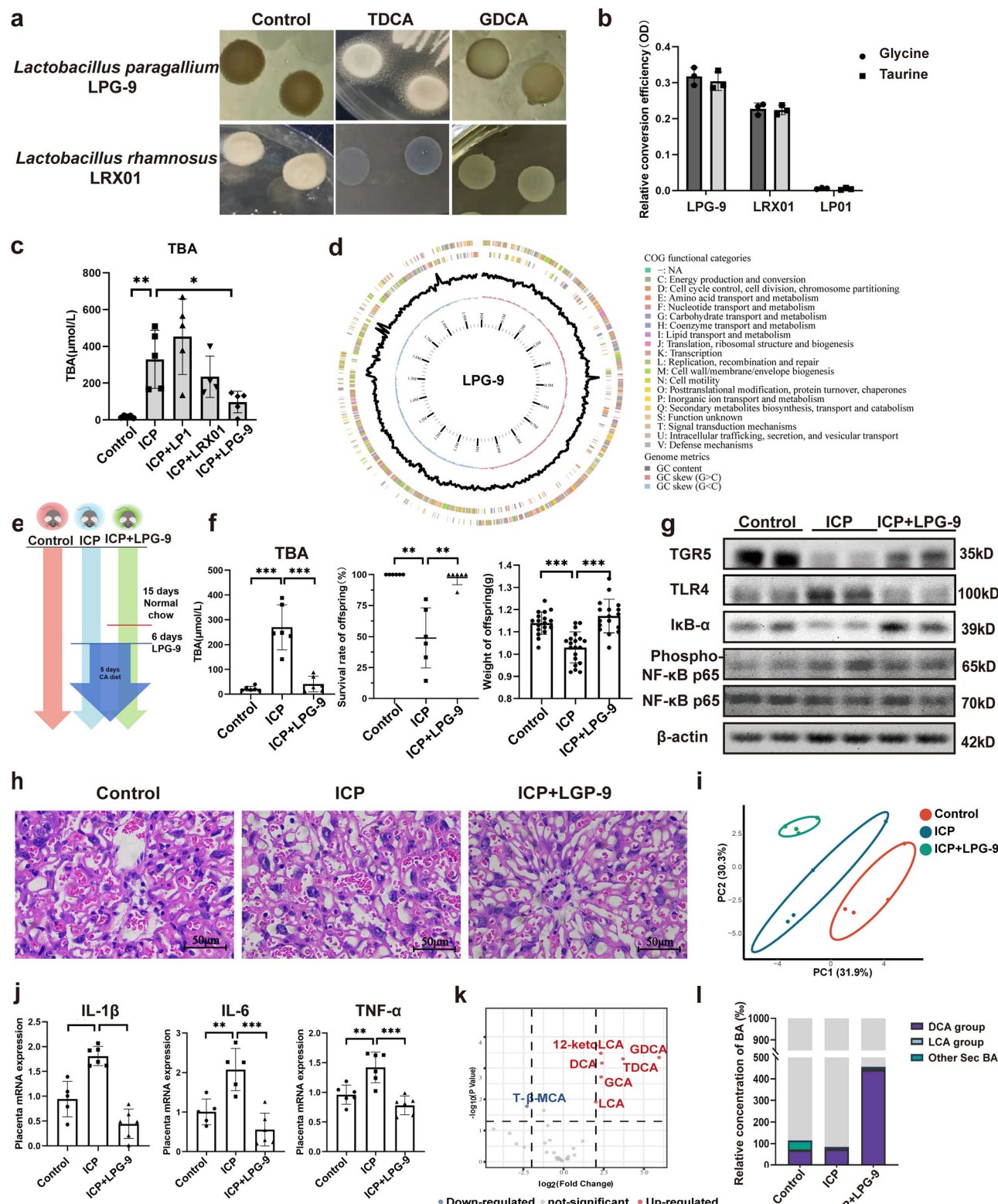

after LPG-9 treatment (Fig. 5c). Moreover, *bsh* expression in cecal contents of the three groups was significantly elevated after LPG-9 treatment (Fig. 5d).

Total fecal bile acids also increased following LPG-9 treatment, consistent with enhanced bile acid excretion (Fig. 5e). Similar to serum findings, the proportion of fecal secondary bile acids significantly increased after LPG-9 treatment, with DCA showing the greatest improvement (Fig. 5f, g). Gene expression analysis of bile acid primary regulators showed that LPG-9 increased hepatic *Bsep* mRNA expression, whereas

ileal *Asbt* and *Ost-α/β* remained unchanged (Fig. 5h), indicating promotion of bile acid secretion via the liver. Additionally, expression of *Fxr*, a key bile acid-regulating receptor, was increased in the livers of LPG-9-treated mice (Fig. 5i). Activation of *Fxr* by LPG-9 was further supported by reduced serum T-β-MCA levels (Fig. 5j). Collectively, these results indicate that LPG-9 alleviates placental inflammation in ICP by enhancing gut bacterial BSH activity to restore the gut microbiota and promote bile acid excretion.

**Fig. 4 |** *Lactobacillus* **strain with BSH and decreased serum TBA levels was obtained. a** Manifestation of bile salt hydrolase activity (BSH) activity by *lactobacilli* on MRS agar medium; **b** Bile salt deconjugation activity of *lactobacilli* (*n* = 3 biologically independent samples; one-way ANOVA); **c** Serum TBA levels in the cholestasis mouse model (*n* = 5 biologically independent samples; one-way ANOVA); **d** Circular genome map of LPG-9. The complete genome is a closed circular chromosome of 1,968,772 bp (GC content 34.87%). From the innermost to outermost circles, the plot displays: a genomic scale indicating the chromosome size and gene positions; the GC skew (black curve), with positive and negative transitions marking the replication origin (oriC) and terminus (ter); the GC content (blue curve), showing fluctuations across the genome; and coding sequences (CDS) on the forward and reverse strands, colored by COG functional categories; **e** Schematic representation of LPG-9 treatment; **f** Serum TBA levels, survival rates and weight of offspring after LPG-9 treatment (*n* = 6 biologically independent samples; one-way ANOVA); **g** Western blotting of TGR5, TLR4 and NF-κB in the mice placenta; **h** Hematoxylin-eosin-stained sections of the placenta after LPG-9 treatment, 400×. Scale bar = 50 μm; **i** Principal component analysis diagram of the serum bile acid; **j** qPCR data shows the gene expression levels of placental pro-inflammatory cytokines (*n* = 5–6 biologically independent samples; one-way ANOVA); **k** Volcano plot of serum bile acids (ICP + LPG-9 vs. ICP) (*n* = 4 biologically independent samples); **l** Stacked bar graph of serum secondary bile acids. Error bars represent standard deviation. Significance levels: *$p < 0.05$, **$p < 0.01$, ***$p < 0.001$.

## Discussion

ICP is characterized by elevated maternal serum bile acids and fetal complications. Here, we found that a reduced proportion of secondary bile acids in ICP serum inhibits TGR5 signaling while activating the TLR4-NF-κB pathway in the placenta. This imbalance is associated with gut microbiota dysbiosis, manifested by a significant reduction in BSH-enriched bacteria, particularly *Lactobacillus*. To validate the role of BSH in the gut microbiota, we fed ICP model animals with LPG-9, which expresses abundant BSH. LPG-9 enhanced bile acid excretion by restoring BSEP and inhibited placental inflammation through TGR5 activation mediated by secondary bile acids.

For every 1-μM increase in maternal bile acids, the incidence of fetal complications increases by 1–2%[16–18], implicating a causal relationship between maternal bile acid levels and fetal complications. Consistent with the findings of previous studies[5,19], we observed elevated TBA levels and increased placental neutrophils in patients with ICP and animal models, accompanied by upregulation of the TLR4-NF-κB pathway and downregulation of TGR5 expression. These findings implicate bile acids as initiators of placental inflammation. NF-κB activation promotes neutrophil chemotaxis and activation[20,21] and prolongs their lifespan by inhibiting apoptosis[22]. The accumulation of neutrophils exacerbates the local inflammatory environment, impairing placental structure and fetal development[14]. This inflammatory response provides a mechanistic link between elevated maternal bile acids and adverse fetal outcomes.

TGR5, which exerts anti-inflammatory effects, is primarily activated by secondary bile acids[23], and its overexpression can effectively inhibit TLR4-NF-κB signaling and the release of downstream cytokines[8,24,25]. The proportion of secondary bile acids in ICP mice serum was reduced, consistent with previous reports demonstrating the predominance of primary bile acids in patients with ICP[26,27], and their elevation in amniotic fluid[28]. This alteration in the bile acid profile impairs TGR5-mediated suppression of TLR4, thereby driving pathological processes. Notably, TGR5 retains functional competence, supported by its inherent GPCR amplification effect[29,30] and unique signaling properties[31]. Our research further confirmed that the highly selective agonist CCDC[32] activated this remaining TGR5, inducing potent anti-inflammatory responses. The placental environment further reinforces the specificity of this intervention, given that other bile acid receptors (FXR, PXR, CAR) exhibit both minimal expression[33] and unchanged expression upon CCDC treatment (Supplementary Fig. 3e). Collectively, these findings emphasize the critical role of TGR5 in ICP and suggest that therapeutic activation of TGR5 through elevated endogenous ligand levels may serve as an effective strategy.

Bile acid composition is closely linked to gut microbiota makeup[34,35]. Alterations in the gut microbiota have been reported in liver diseases such as cholestatic liver disease[36], alcoholic liver disease, and cirrhosis[37,38]. In our study, the composition of the intestinal bacteria of ICP mice was markedly altered, characterized by a decreased abundance of *Lactobacillus*. This reduction was closely associated with reduced fecal BSH activity. We propose that the characteristic systemic elevation of bile acid levels in ICP is the primary driver of these microbial community changes. The bactericidal effect of elevated bile acids is well established. Gram-positive bacteria are susceptible to bile acid damage due to their relatively simple cell membranes[39], which are permeable to small molecules. The outer membrane of gram-negative bacteria comprises anionic lipopolysaccharides, which restrict the penetration of hydrophobic substances such as bile acids[40,41]. In Apc^min/+ mice, CA decreased the abundance of *Lactobacillus*[42]. Conversely, inhibition of endogenous bile acids synthesis significantly increased the abundance of intestinal gram-positive bacteria, including *Lactobacillus casei* and *Lacticaseibacillus paracasei*[43]. Notably, BSH activity is found predominantly in gram-positive bacteria in mice and humans. Among the BSH phylotypes, the most active BSH-T3 exists only in *Lactobacillus*[44]. This indicates that *Lactobacillus*, as gram-positive bacteria, are sensitive to elevated bile acid levels, resulting in reduced abundance of *Lactobacillus* and BSH-enriched gut microbiota in ICP feces.

BSH exists in the gut microbiome and catalyzes the hydrolysis of conjugated bile acids to unconjugated forms, influencing various diseases. This is supported by the administration of *Parabacteroides distasonis*, which reduced bile acids in the liver and ameliorated hepatic fibrosis by increasing BSH activity[45]. Similarly, in cholestatic drug-induced liver injury, increasing the abundance of BSH-enriched intestinal commensal bacteria promotes bile acid excretion[46]. Conversely, the loss of these bacteria is detrimental, as evidenced by the significantly reduced fecal abundance of BSH-enriched bacteria in cholestatic preterm infants[47]. Furthermore, long-term supplementation with exogenous bile acids in mice reduced the abundance of BSH-enriched bacteria and induced inflammation[48]. We transplanted fecal microbiota from pregnant women with ICP into normal pregnant mice and observed mild ICP-related symptoms and placental inflammation. These observations are consistent with previous studies[49,50], and directly confirm the pathogenic potential of the gut microbiota in ICP patients. This pathogenicity is closely associated with the critical functional defect of reduced BSH activity. Based on this, we propose that the dysregulation of both gut microbiota and bile acids in ICP triggers a self-reinforcing vicious cycle. ICP-induced elevation in bile acid levels selectively inhibits the growth of gram-positive bacteria, particularly BSH-enriched strains, altering bile acid metabolism and exacerbating systemic bile acid load and its associated toxic effects. Therefore, given that BSH deficiency is linked to both gut microbiota dysbiosis and bile acid disorders, *Lactobacillus* strains with BSH genes that regulate bile acid represent an effective intervention option for placental inflammation in ICP.

Using whole-genome sequencing, we identified LPG-9, a BSH-enriched *Lactobacillus paragasseri* strain, and confirmed its safety, probiotic traits, and high tolerance to gastric fluid and bile salts[51], which favor its survival within the high bile acid environment of patients with ICP. LPG-9 intervention significantly increased the relative abundance of *Lactobacillus* and enhanced intestinal BSH activity, indicating its successful colonization and functional benefit. Specifically, through its BSH activity, LPG-9 hydrolyzed conjugated primary bile acids in the gut into unconjugated forms, thereby facilitating their subsequent microbial conversion into secondary bile acids. This alteration significantly elevated secondary bile acid levels and reduced the concentration of the FXR antagonist T-β-MCA. These bile acids exert therapeutic effects through two pathways. In the placenta, secondary bile acids acted as endogenous agonists of the TGR5 receptor, whose activation potently inhibited the TLR4-NF-κB signaling pathway, thereby directly mitigating inflammation. Concurrently, in the liver, the decreased level of T-β-MCA alleviated FXR inhibition, resulting in significant upregulation of FXR and its downstream transporter BSEP. This

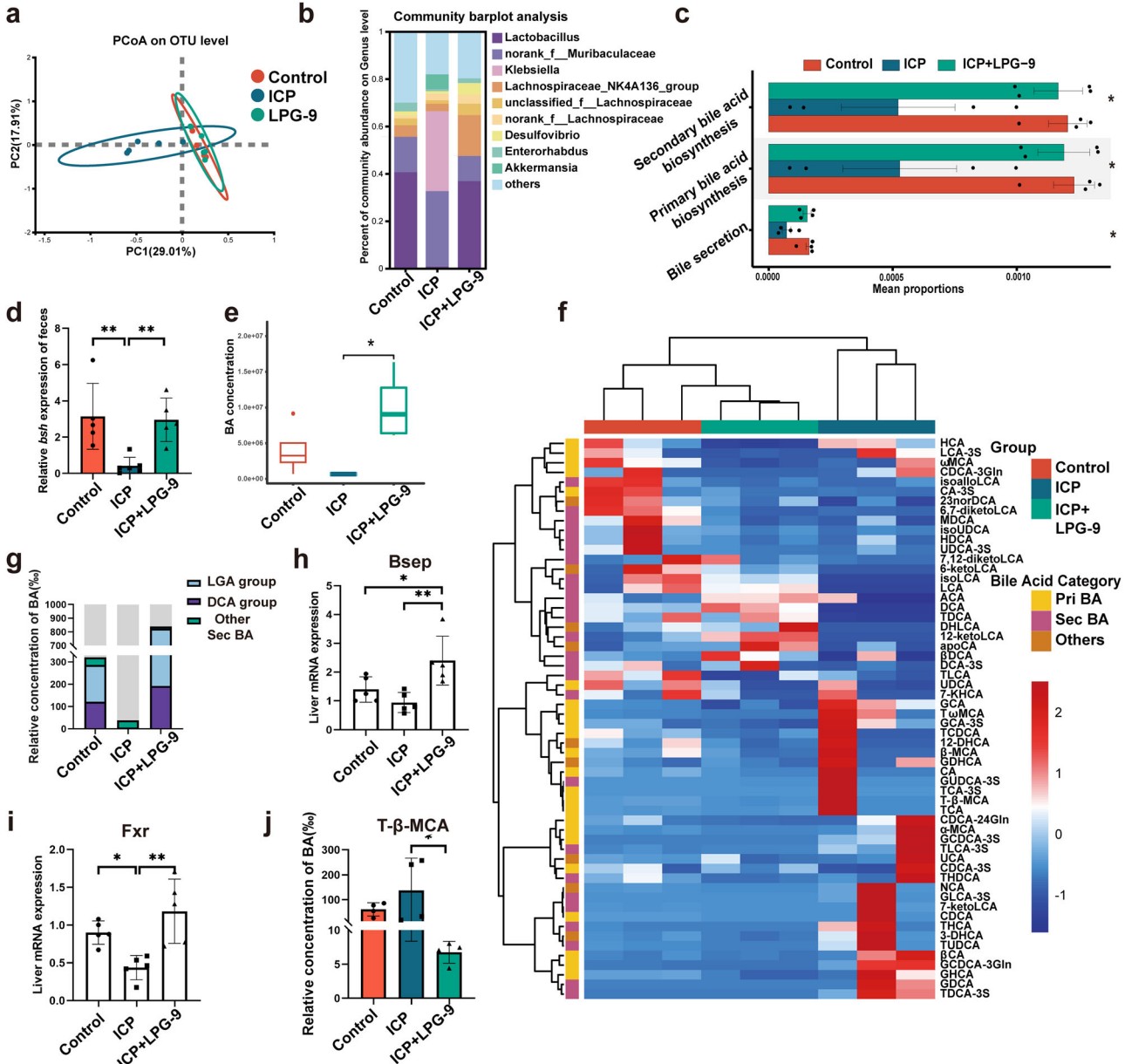

**Fig. 5 | LPG-9 restores gut microbiome balance in ICP and promotes bile acid secretion. a** Principal coordinate analysis diagram of gut microbiome; **b** Bar diagram of the fecal microbial composition; **c** KEGG pathway prediction results based on Tax4fun. Error bars represent standard error; **d** *bsh* expression in feces (*n* = 5–6 biologically independent samples; one-way ANOVA). Error bars represent standard deviation; **e** Concentration of the total fecal bile acids. Error bars represent standard deviation; **f** Heat map of fecal bile acid. The color scale represents normalized abundance (row Z-score), with blue indicating low abundance and red indicating high abundance; **g** Stacked bar graph of secondary bile acids in feces. Error bars represent standard deviation; **h** Gene expression levels of bile salt export pump (*Bsep*) (*n* = 5–6 biologically independent samples; one-way ANOVA). Error bars represent standard deviation; **i** Gene expression levels of hepatic *Fxr* (*n* = 5–6 biologically independent samples; one-way ANOVA). Error bars represent standard deviation; **j** Concentration of T-β-MCA (FXR inhibitor bile acid) (*n* = 4 biologically independent samples; one-way ANOVA). Error bars represent standard deviation. Significance levels: *$p$ < 0.05, **$p$ < 0.01, ***$p$ < 0.001.

promoted bile acid secretion into the bile ducts and subsequently reduced serum TBA levels. These findings are consistent with the observations that BSEP downregulation contributes to ICP pathogenesis[52], its activation alleviates cholestatic liver injury[53], and FXR signaling exerts a protective role[54,55]. Therefore, LPG-9 inhibited placental inflammation through secondary bile acid-mediated TGR5 activation and promoted hepatic bile acid excretion via BSEP, providing a dual-target probiotic therapy for ICP (Fig. 6).

However, supplementation with BSH-active *Bacteroides fragilis* has been reported to not be beneficial in ICP[50]. Probiotics of different BSH subtypes differ significantly in terms of bile acid catalysis efficiency and substrate preference[56]. Additionally, in vivo experiments have revealed that

probiotics of different BSH subtypes produce different major metabolites[57], suggesting that the effect of BSH enzymes on ICP varies depending on the enzyme source.

This study had two limitations. First, the clinical sample size was limited; thus, more clinical data are required for further analysis. Second, the specific function of LPG-9 BSH compared with that of other BSH-active bacteria remains unclear, necessitating further investigations.

## Conclusions

Our results revealed that gut microbiota dysbiosis, characterized by a reduction in BSH-active *Lactobacillus*, disrupts bile acid homeostasis, triggering placental inflammation via the TGR5-TLR4-NF-κB axis.

**Fig. 6 | Mechanism by which LPG-9 improves placental inflammation.** LPG-9 modified the bile acids composition by converting primary bile acids (Pri BAs) to secondary bile acids (Sec BAs) via BSH. Among these, placental TGR5, activated by secondary bile acids, negatively regulated the TLR4-NF-κB signaling pathway and attenuated ICP-associated placental inflammation. Reduced T-β-MCA activated the hepatic FXR-BSEP signaling pathway and increased hepatic bile acid secretion. No significant change in bile acid uptake was observed, thus promoting bile acid excretion in ICP model mice.

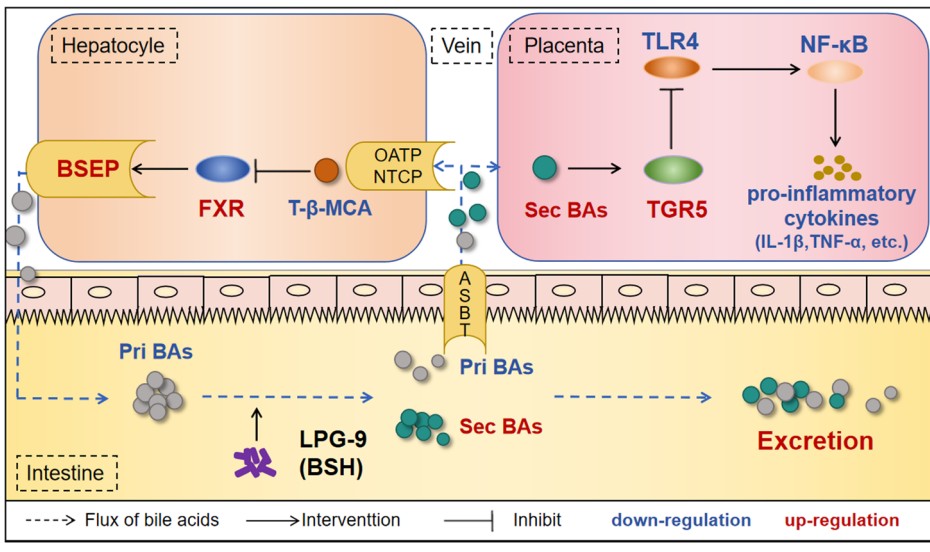

Additionally, we demonstrated that intervention with BSH-enriched LPG-9 represents a dual-target strategy for ICP, which simultaneously activated placental TGR5 to inhibit TLR4-NF-κB signaling and promoted hepatic bile acid excretion via BSEP. These findings not only reveal a critical mechanism of host-microbiome interactions in the pathogenesis of ICP, but also provide a transformative framework for developing microbiome-based therapies to improve pregnancy outcomes.

## Materials and methods
### Participants and sample collection
This retrospective study (NFEC-2021-054) was approved by the Ethics Committee of Nanfang Hospital, Guangzhou, China. Placentas were collected from five healthy mothers and five patients with ICP immediately after caesarean or vaginal delivery and stored at −80 °C. Informed consent was obtained from all participants prior to sample collection. All samples were rigorously de-identified prior to analysis, ensuring that no personal identifying information could be linked to the samples. All ethical regulations relevant to human research participants were followed.

### Bacterial and cell cultures
*Lactobacillus paragasseri* LPG-9 and *Lactiplantibacillus plantarum* LP1 were originally isolated from the gastric mucosa of healthy individuals, while *Lacticaseibacillus rhamnosus* LRX01 was isolated from the feces of newborns. The bacterial strains were deposited at the Guangdong Institute of Microbiology, China. LPG-9, LP1, and LRX01 were cultured aerobically for 24 h at 37 °C in MRS broth (Huankai, Guangzhou, China). *J774.2* and *HEK293T* cells were obtained from ATCC (Manassas, VA, USA). The cells were authenticated by the supplier and tested negative for mycoplasma contamination before use. The cells were cultured in DMEM supplemented with 10% fetal bovine serum (GIBCO, Grand Island, NY, USA) and maintained at 37 °C in a 5% $CO_2$ atmosphere.

### Animals
C57BL/6 female mice were obtained from the Southern Medical University Experimental Animal Center. All animal experiments were approved by the Ethics Committee of Southern Medical University (permit NO. 44002100006397). We have complied with all relevant ethical regulations for animal use. The sample size was calculated using the resource equation method ($n = E/K + 1$)[58] combined with a priori using a power analysis. All housing and treatment procedures followed ARRIVE guidelines. Mice were housed under standard specific-pathogen-free (SPF) conditions with a 12-h light/dark cycle. Cages were randomly placed within the animal facility to minimize location-related confounding. Female virgin mice (8–12 weeks old) were paired with fertile males, and the presence of a vaginal plug was

designated as gestational day (GD) 0. Pregnant C57BL/6 mice were fed *ad libitum* either a normal chow (NC) or a cholic acid (CA) diet supplemented with 0.5% or 1% (w/w) CA from GD16 until delivery ($n = 6$ per group), following an established protocol[59]. Successful ICP modeling was confirmed by maternal serum TBA levels and histopathological changes in liver tissue. Mice that failed to develop ICP were excluded. Data on offspring survival and body weight were collected after euthanasia Blood, liver tissue, and fecal samples were collected for further analysis.

For the TGR5 receptor agonist CCDC (3-(2-chlorophenyl)-N-(4-chlorophenyl)-N,5-dimethyl-4-isoxazolecarboxamide) treatment, pregnant mice were fed a 0.5% (w/w) CA diet and 5% (w/w) CCDC (HY-14229, MCE, USA) dissolved in water from GD16 until delivery.

FMT was performed using a modified protocol adopted in previous studies[60]. Fecal donors included three randomly selected patients with ICP. To prepare microbiota suspensions, fecal samples (2–3 g) were washed three times with sterile PBS, mixed with PBS containing 20% (w/v) glycerol, filtered, and stored at −80 °C until use. Recipient mice were orally administered the prepared fecal suspensions daily from GD14 until delivery. During the experiment, all mice were fed an NC diet.

For probiotic therapy, LPG-9 was washed via centrifugation in sterile PBS for probiotic therapy. Pregnant mice were orally administered $2 \times 10^9$ CFU d⁻¹ of LPG-9 for 7 days, starting from GD14, followed by induction with a 0.5% CA diet.

### Cell model
Mouse macrophage *J774.2* cells were divided into three groups. In the lipopolysaccharide (LPS) group, 0.5 mg mL⁻¹ LPS (L8880, Solarbio, China) was added, gently mixed, and incubated for 1 h. In the LPS + CCDC group, 3 µM TGR5 receptor agonist CCDC (HY-14229, MCE, USA) was added 2 h before LPS. After incubation, the medium was removed, and cellular proteins were extracted.

### Serum marker testing
After euthanasia, serum samples were centrifuged (4 °C, 3000 rpm, 15 min) and stored at −80 °C until further analysis. Levels of TBA, AST, and ALT were measured using an automatic biochemical analyzer (AU2700, Olympus, Japan).

### 16S rRNA sequencing of microbial DNA and microbial community analysis
16S rRNA amplicon sequencing and subsequent analyses were performed by MAGIGENE (Guangzhou, China). Genomic DNA was extracted from cecal samples with a bacterial DNA extraction kit (TaKaRa Bio), and quality was verified using 1% agarose gel electrophoresis. The V3−V4 hypervariable

region of the bacterial 16S rRNA gene was amplified with primers 338F and 806 R. After quantification, normalization, and pooling, libraries were sequenced using MiSeq v3 reagents. Bioinformatic processing was performed using QIIME (V1.9.1). Chimeric sequences were removed using USEARCH, and operational taxonomic units were clustered at 97% similarity using UPARSE. Taxonomic assignment was performed using the Database Project (RDP) classifier against the GenBank database at a confidence threshold of 0.7. Principal coordinate analysis (PCoA), principal component analysis (PCA) charts, and heatmaps were generated with the R software (v3.5.0). Group differences were identified using linear discriminant analysis effect size analysis (LEfSe) with an LDA score of ≥3.5. Gut microbiome functions were predicted based on 16S rRNA sequencing data using Tax4Fun and KEGG functional annotation.

## Bile acid-targeted metabolomics

Bile acids were extracted from fecal and serum samples using a targeted metabolomics protocol[61]. Briefly, sample preparation was conducted using HPLC (high-performance liquid chromatography)-grade solvents supplemented with 250 nmol deuterated bile acid standard. After centrifugation, drying, and resuspension, the homogenized solution was filtered for HPLC-mass spectrometry (HPLC-MS) using the Q Exactive Focus Thermo Fisher Scientific/1290 Infinity series UHPLC System (Agilent, USA).

## RNA extraction and quantitative real-time PCR

Total RNA from each group was extracted using a TRIzol reagent (9108, TaKaRa Bio, Japan), according to the manufacturer's instructions. All RNA samples were validated for purity via ultra-micro spectrophotometer, with A260/A280 ratios ranging between 1.9 and 2.1, and A260/A230 ratios consistently exceeding 2.0. RNA integrity was assessed by agarose gel electrophoresis. Total RNA was separated on a 1% agarose gel, and the presence of intact, sharp 28S and 18S ribosomal RNA bands (Supplementary Fig. 5). RNA reverse transcribed into cDNA using a reverse transcription kit (RR047A, TaKaRa Bio, Japan). Gene expression was detected using SYBR Green premix on a 7500 real-time PCR system (Applied Biosystems, USA), according to the SYBR Green Universal PCR Premix protocol. Each sample was analyzed in triplicate, and levels were normalized to the control housekeeping gene GAPDH. Relative quantitative analysis of mRNA was performed using the comparative threshold cycle method, $2^{-\Delta\Delta CT}$, with extreme outliers excluded. The primer sequences are listed in Supplementary Table 1.

## Co-immunoprecipitation (Co-IP)

For Co-IP of native proteins, the samples were completely homogenized with a lysis buffer (HY-K0202K-B, MCE, USA), the samples were centrifuged for 10 min at 12,000 rpm and 4 °C, and collected at −80 °C for subsequent testing. Next 500 μg of cell lysate was incubated overnight at 4 °C with mouse anti-TLR4 (1:100; 66350, Proteintech, USA) and rabbit anti-TGR5 (1:50; A20778, ABclonal, China) antibodies, using IgG (1:1000) as a negative control. Subsequently, the complexes were incubated with magnetic beads (HY-K0202K-A, MCE, USA) for 1 h at 25 °C. After collecting the magnetic beads and washing them thrice with PBS, the remaining beads were resuspended in a loading buffer and subsequently analyzed through western blotting using anti-TGR5 antibody and anti-TLR4 antibodies.

For Co-IP of overexpressed proteins, plasmids encoding N-terminally HA-tagged TGR5 and N-terminally FLAG-tagged TLR4 were constructed and co-transfected in *HEK293T* cells using lipo8000 (C0533, Beyotime, China). After 24 h, cells were lysed in a non-denaturing IP lysis buffer (HY-K0202K, MCE, USA). The cell lysate was incubated with anti-FLAG (HY-K0207, MCE, USA) and anti-HA magnetic beads (HY-K0201, MCE, USA) to immunoprecipitate FLAG-TLR4 and anti-HA, respectively, overnight at 4 °C. After collecting the magnetic beads and washing them thrice with PBS, the remaining beads were resuspended in a loading buffer and subsequently analyzed via western blotting using anti-HA (1:5000) and anti-FLAG (1:1000) antibodies.

## Western blotting

The proteins were separated using 12.5% polyacrylamide gel electrophoresis and transferred to PVDF membranes. Due to the different molecular weights of the target proteins, membranes were cut into strips to probe multiple targets from a single gel. After blocking with 3% BSA in TBST for 1 h, the membranes were incubated with anti-TLR4 mouse (1:1000; 66350, Proteintech, USA), anti-TGR5 rabbit (1:100; A20778, ABclonal, China), anti-phospho-NF-κB p65 rabbit (1:1000; 3033S, Cell Signaling, USA), anti-NF-κB p65 rabbit (1:1000; AB32536, Abcam, USA), anti-TNF-α rabbit (1:1000; 17590, Proteintech, China), anti-IL-6 rabbit (1:1000; 21865, Proteintech, China), anti-IL-1β rabbit (1:1000; 16806, Proteintech, China), anti-HA (1:5000; R20003, Abmart, China) and anti-FLAG (1:1000; AF519, Beyotime, China) antibodies overnight at 4 °C. This was followed by incubation with a secondary antibody (1:5000; SE134, Solarbio, China. 1:5000; SE131, Solarbio, China) for 1 h. Blotting assays were visualized using ECL detection reagents (1705061, Bio-Rad, USA). Additional non-specific bands visible in some lanes (labeled "NS" in the original image) may result from the polyclonal nature of antibodies and the complexity of placental tissue lysates. These bands were excluded from the main figure panel. All original blot images are included in Supplementary Fig. 6.

## Hematoxylin-eosin (HE) staining and immunohistochemical (IHC) staining

Fresh placental tissues were collected and fixed in 10% formalin. For HE staining, samples were embedded in paraffin, sectioned, and stained, following standard protocols. For immunohistochemical staining, placental tissue sections were incubated overnight at 4 °C with antibodies specific for MPO, TGR5, TLR4 and phospho-NF-κB p65, followed by a 50-min treatment with appropriate peroxidase-conjugated secondary antibodies at room temperature and diaminobenzidine development. Subsequently, the sections were stained with hematoxylin for 3 min. All images were captured using a digital microscope camera (Nikon, Tokyo, Japan), and IHC images were quantified using the Image-Pro Plus software (Media Cybernetics, MD, USA).

## Determination of bile salt hydrolase (BSH) activity

BSH activity was analyzed based on a previously reported method[61]. Bacteria ($2 \times 10^9$ CFU) were collected via centrifugation and resuspended in PBS. Subsequently, they were broken for 30 min using an ultrasonic cell crusher (2:3, 500 W). Next, 10 μL of the suspension was spotted onto the MRS agar containing taurodeoxycholic acid (5 g L$^{-1}$; T835753, Macklin, China) or glycodeoxycholic acid (5 mM; S817460, Macklin, China), followed by incubation for 72 h at 37 °C. BSH activity was confirmed by the formation of a white precipitate or halo.

For quantitative analysis, bacteria ($2 \times 10^9$ CFU) were collected and lysed. After centrifugting the mixture (4 °C, 10,000 rpm, 15 min), 100 μL of the supernatant was combined with 1.8 mL of PBS and 100 μL of taurodeoxycholic acid or glycodeoxycholic acid (200 mM). After a 30-min incubation at 37 °C, 15% trichloroacetic acid (w/v) (500 μL) was added, followed by centrifugation (4 °C, 14,000 rpm, 10 min). Furthermore, the resulting supernatant (100 μL) was reacted with 1.9 mL of ninhydrin chromogenic solution in a boiling water bath for 14 min, and the absorbance was measured at 570 nm to quantify enzymatic activity.

## Statistics and reproducibility

For animal studies, sample size was calculated using the resource equation method combined with a priori power analysis. Mice were randomly assigned to experimental groups based on gestational order, and cages were randomly placed within the animal facility. Throughout the experiment, researchers responsible for tissue collection and histological analysis were blinded to group assignments. All experiments were performed with at least three biologically independent samples.

Statistical analyses between groups were performed using the Wilcoxon rank-sum test, t-test, or one-way ANOVA, depending on the data. Correlation analyses were performed based on Spearman's correlation

coefficients. Statistical significance was set at $p < 0.05$ (*$p < 0.05$, **$p < 0.01$, and ***$p < 0.001$). IBM SPSS (version 22.0; IBM SPSS, USA), GraphPad Prism version 8.0, and the R package (3.5.0) were used for statistical analysis and data visualization. Exact $p$ values, effect sizes, and confidence intervals for all statistical tests are provided in the Source Data file (Supplementary Data 1).

## Data availability

The raw sequencing data generated in this study have been deposited in the NCBI Sequence Read Archive (SRA) under BioProject accession number PRJNA1238783. The newly generated plasmids, pCDNA3.1(-)-TLR4-FLAG and pCDNA3.1(-)-TGR5-HA, have been deposited in the WeKwikGene plasmid repository under accession IDs 0002377 and 0002378, respectively. Requests for these plasmids should be directed to the corresponding author or the WeKwikGene repository. The original protein blotting images are provided in Supplementary Fig. 6, and the source data for all charts presented in the article are available in Supplementary Data 1. All other data supporting the findings of this study are available from the corresponding author upon reasonable request.

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

## Acknowledgements
This study was supported by the National Natural Science Foundation of China [No. 32370139 and 32300085], Key-Areas Research and Development Programs of Guangdong Province [No.2022B1111070006], and Guangdong Basic and Applied Basic Research Foundation [No. 2023A1515012538 and 2025A1515010567].

## Author contributions
Conceptualization: H. Fan, W. Huang and Q. Lin; Data curation and formal analysis: W. Zeng, W. Huang, J. Zhang and Q. Lin; Investigation & Validation: J. Shan, W. Shen, P. Du, J. Zhang, J. Liu and X. Guo; Writing-original draft: W. Huang and Q. Lin; Writing-review and editing: H. Fan, W. Huang, Q. Lin and Z. Chen; Supervision: H. Fan.

## Competing interests
The authors declare no competing interests.
