## [Transparent Peer Review file · Communications Biology]

Lactobacillus paragasseri LPG-9 Reduces Placental Inflammation in Intrahepatic Cholestasis of Pregnancy by Regulating TGR5 in mice

Corresponding Author: Professor Hongying Fan

Version 0:

Reviewer comments:

Reviewer #1

(Remarks to the Author)

The manuscript entitled “L. paragallium LPG-9 reduces placental inflammation in intrahepatic cholestasis of pregnancy by regulating TGR5” proposes that TLR4-mediated NF- κ B signaling is elevated by the decrease of TGR5 in the placenta of a patient with ICP, based on the first evidence from a single human case. The authors then established an ICP mouse model and indicated that the BSH-producing bacteria were reduced. Using faecal microbiota transplantation and L. paragallium LPG-9 supplement, they concluded that L. spp were altered in ICP and played an essential role in placental inflammation through secondary BAs. Overall, this study is a significant contribution for the inflammation of the placental, connecting BA metabolism through intestinal microbiota with placental inflammation in ICP patients. However, there are several minor points that need to be addressed.

1. There is only one patient sample. The authors need to address why only one case was applied for the study. For the immunostainings, high resolution pictures are helpful to determine true changes of TGR5, TLR4 and NF- κ B. Which subunit of NF- κ B was measured in the experiment? The scale bars are not clear in the pictures.
2. For ICP mouse model, evidences were required for a successful model of ICP from the liver using staining and other helpful biomarkers.
3. Since the authors proposed that it is neutrophil that contributes to the activated inflammation in ICP, they need to show evidence to support the rationale in neutrophils or explain why.
4. If possible, please provide better Western Blot results. In some cases, it is hard to conclude the changes as proposed by the authors, such as Fig1. J, K.
5. Please provide in vitro colP to support the protein-protein interaction between TLR4 and TGR5, using different combinations.
6. In Fig.2J, 8 samples were limited to show the correlation between BSH and L. spp. The experimental number should be expanded.
7. Discussion is necessary for the reason for the reduction of L. bacteria in the intestine of ICP patients.

Reviewer #2

(Remarks to the Author)

Lactobacillus paragallium LPG-9 Reduces Placental Inflammation in Intrahepatic Cholestasis of Pregnancy by regulating TGR5

By Huang, et al.

The study explores the interplay between the gut microbiome, bile acid metabolism, and placental inflammation associated with Intrahepatic cholestasis of pregnancy (ICP). ICP involves elevated bile acids and placental inflammation linked to reduced TGR5 activity. The study aims to use probiotics to help mitigate ICP-associated placental inflammation.

This study is significant because cholestasis affects approximately 0.1% of pregnancies and is linked to increased risks of stillbirth, preterm labor, and complications in mothers as well. Overall, the paper is scientifically sound, study design is reasonable, and analyses are appropriate to address the hypothesis. The manuscript is clear and maintains a logical flow and cohesiveness. The following suggestions will improve the manuscript quality and readability:

Comments:

- 1- There are awkward phrasing and incorrect word use as well as grammatical and typographical errors throughout the manuscript. Editing for grammar and syntax is needed.
- 2- L77: conclude the Introduction with a statement of how the planned work is expected to advance the field.
- 3- Line 193, use a consistent bacterial name for LPG-9, is it *Lactobacillus paragasseri* or *Lactobacillus paragallium*?
- 4- L215-226: Figure panels of J-L are missing.
- 5- Discussion: The discussion generally reiterates the results without much additional insight into interpretation or context for the study. Needs significant improvement.
- 6- L248-251: The reported association between LPG-9 supplementation and increased hepatic bile acid secretion is a significant finding that needs further discussion. Proposing a potential mechanistic pathway would strengthen the interpretation of this result.
- 7- L268: correct "signaling"
- 8- Line 318: Use full name for *Parabacteroides distasonis*
- 9- Line 342: Use full name for *Bacteroides fragilis*
- 10- Lines 378–380: Update the genus names for *Lactobacillus plantarum* and *Lactobacillus rhamnosus* to their current genus classifications, *Lactiplantibacillus plantarum* and *Lacticaseibacillus rhamnosus*, respectively. Please ensure these changes are applied consistently throughout the manuscript if the outdated names appear elsewhere.
- 11- Data Analysis: Were the number of human placenta samples and number of animals per group were chosen based on power analysis? There is no info on the power analysis and no indication what outcome the study was powered or what is the primary outcome of the current study. The parameters (primary outcomes, and secondary outcomes if appropriate) used to estimate study power must be included.
- 12- qPCR analysis: What were the RNA integrity quality numbers for qPCR analyses? This helps to see if the integrity was high enough to get reliable qPCR results.
- 13- Conclusion is a restatement of results without any integration or advancement in knowledge.

Version 1:

Reviewer comments:

Reviewer #1

(Remarks to the Author)

Major points:

1. The authors present that fecal matter from ICP patients can induce the condition (Fig. 3), while *Lactobacillus paragallinarum* LPG-9, potentially sourced from a healthy individual, appears to alleviate it (Fig. 4). These findings seem to present a significant conflict with the central hypothesis. Could the authors please clarify this apparent discrepancy or further discuss the scope and limitations of these results?
2. The authors report that TGR5 expression is decreased in ICP and that its agonist reduces inflammatory signaling. This raises a mechanistic question: how can an agonist be effective if the target receptor is downregulated? The authors should clarify the proposed mechanism. For instance, is the residual TGR5 expression sufficient for agonist-mediated signaling, or does the agonist work through an alternative pathway? This point should be addressed to strengthen the study's conclusions.

Minor points:

1. Since the clinical data from the five patients form the foundation of this study, it is crucial to present the key staining and Western blot results for all individual cases. Showing the complete dataset would strengthen the manuscript by demonstrating the reproducibility and consistency of the findings across the patient cohort.
2. For improved clarity and precision, we suggest that the authors revise the statement on line 131 to explicitly state that there was a decrease in **Lactobacillus** in the ICP group.
3. In Figure 2H, the label for one of the microbes appears to be overlapped, which makes it difficult to read. Please correct the label to ensure all text is clearly legible.

Reviewer #2

(Remarks to the Author)

Thank you for adequately addressing all my comments and suggestions. I only recommend that you include your description of the RNA quality testing in the Methods section of the manuscript. The RNA quality figure can be placed in the supplementary material file.

Reviewer #1

Dear reviewer:

We sincerely appreciate you taking the time to review our manuscript and providing such valuable and constructive feedback. Your insightful suggestions have been instrumental in enhancing the robustness and completeness of this study. We have carefully addressed each point raised and have thoroughly revised and supplemented the manuscript accordingly. We believe the revised manuscript has been substantially improved and provides stronger support for our scientific conclusions. Our point-by-point responses are detailed below for your consideration.

Comment 1: *There is only one patient sample. The authors need to address why only one case was applied for the study. For the immunostainings, high resolution pictures are helpful to determine true changes of TGR5, TLR4 and NF- κ B. Which subunit of NF- κ B was measured in the experiment? The scale bars are not clear in the pictures.*

Response 1: Thank you for raising this important point regarding the clinical samples. We would like to clarify that the analysis of human placental tissue in this study was not based on only one patient sample, but rather on five patient sample per group. This information is clearly stated in the Materials and methods section of the revised manuscript (lines 381–383) and in the legend of manuscript Fig. 1.

For the immunostainings, we used a phospho-NF- κ B p65 antibody targeting the NF- κ B p65 subunit, which is the primary transcription activator subunit of the NF- κ B pathway and a classic marker of its activation. This point has been clearly stated in the revised manuscript's Materials and methods section (lines 518) and in the legend for manuscript Fig. 1.

In addition, we have replaced the immunohistochemical images (including manuscript Fig 1, 3, and 4) with new ones that feature a clear scale bar in the revised version. Response Fig. 1 is an example of the updated figure for your review.

Response Fig. 1 The revised immunohistochemical images (200 \times)

Comment 2: *For ICP mouse model, evidences were required for a successful model of ICP from the liver using staining and other helpful biomarkers.*

Response 2: Thank you very much for the valuable suggestions. As described in our manuscript, the ICP model was established by feeding mice a cholic acid diet from gestational day 16 until delivery, following a previous protocol¹. The successful establishment of the model was verified through multiple lines of evidence. Primarily, we observed a significant increase in maternal serum TBA, which is the hallmark

biochemical feature of ICP. Most critically, we have supplemented the manuscript with histopathological evidence from HE staining of the liver. This analysis demonstrated typical features of cholestasis, including hepatocyte ballooning, inflammatory cell infiltration, and the presence of bile plugs, particularly in the 1.0% CA group. The HE staining results of the liver in the ICP model mice are presented in Response Fig. 2 for your review and has been included in the manuscript as revised Fig. S1C (lines 77 – 79). Furthermore, the model recapitulated the typical adverse fetal outcomes associated with ICP, such as reduced offspring survival and body weight.

Taken together, the elevated serum TBA, cholestatic liver histopathology, and adverse fetal outcomes offer compelling proof of a successfully established ICP model.

Response Fig. 2 Hematoxylin-eosin-stained sections of the liver in the ICP model mice (200×). Arrows indicate bile plugs

Comment 3: *Since the authors proposed that it is neutrophil that contributes to the activated inflammation in ICP, they need to show evidence to support the rationale in neutrophils or explain why.*

Response 3: Thank you for raising this critical point. To address this, we performed immunohistochemical staining for myeloperoxidase (MPO) to assess placental neutrophil infiltration. The results showed consistent findings with HE staining, demonstrating a significant increase in neutrophil infiltration in the ICP group compared to the control group. The results we observed in placental MPO staining of ICP model mice were also consistent with the human situation, providing direct histological evidence for neutrophil involvement in ICP-induced inflammation. The results of this experiment are presented in Response Fig. 3 for your review and has been included in the manuscript as revised Fig. S1A and D (lines 63 – 85 and 83 – 85). Based on this finding, we propose the following mechanism pathway: elevated maternal bile acids activate the NF- κ B pathway in placental cells.

It has been demonstrated that NF- κ B activation accelerates neutrophil chemotaxis and activation, while also prolonging their survival by inhibiting apoptotic pathways²⁻⁴. This increases the infiltration of neutrophils into the placenta. Infiltrating activated neutrophils exacerbate inflammation through the release of myeloperoxidase and so on, leading to placental structural damage and impaired fetal development⁵. Furthermore, the population-based study by Luo et al. (2022) and Hasan et al. (2021) revealed a positive correlation between inflammatory responses during pregnancy and neutrophil counts, suggesting that neutrophils may serve as a potential biomarker for ICP^{6,7}.

Therefore, based on the findings of this study (manuscript Fig. 1) and the new histological evidence, we propose that elevated maternal bile acids in ICP triggered placental inflammation by activating the NF- κ B pathway. This information is stated in the Discussion section of the revised manuscript (lines 275 – 281).

Response Fig. 3 Myeloperoxidase-stained sections of the placenta in humans and mice with ICP (200 \times). Arrows indicate neutrophils.

Comment 4: *If possible, please provide better Western Blot results. In some cases, it is hard to conclude the changes as proposed by the authors, such as Fig1. J, K.*

Response 4: Thank you for your valuable comments, which have greatly helped us improve the quality of our paper. We agree that the quality of the original Western blot images in Figures 1J and 1K could be improved. In response, we have repeated the Western blot experiments for these figures.

The new, high-quality blot images are presented below (Response Fig. 4A) and have replaced the original ones in the revised manuscript (Figs. 1J and 1K). As shown, the new blots exhibit sharper bands, lower background, which allow for a clear and reliable quantification of the protein levels. The results from these improved experiments robustly confirm our original conclusions.

As additional supporting evidence, all the key findings from the Western blot analysis in this study were consistently supported by qPCR results at the mRNA level (Response Fig. 4B). We are grateful for the reviewer's suggestion, which has undoubtedly strengthened the quality of our data.

Response Fig. 4 Western blot and qPCR data showing the expression levels of placental TGR5, TLR4, and NF-κB.

Comment 5: Please provide *in vitro* coIP to support the protein-protein interaction between TLR4 and TGR5, using different combinations.

Response 5: We thank the reviewer for this excellent suggestion. In the original manuscript, we provided evidence for the TGR5-TLR4 interaction through Co-IP under endogenous conditions in *J774.2* macrophages (manuscript Fig. 1K, shown here in Response Fig. 5A for your review). To further strengthen this finding, we have now performed Co-IP experiments with overexpressed proteins in *HEK293T* cells, as requested. The results of this experiment are presented in Response Fig. 5B for your review and has been included in the manuscript as revised Fig. S1B (lines 95 – 98).

Briefly, HEK293T cells were co-transfected with plasmids encoding N-terminally HA-tagged TGR5 and N-terminally FLAG-tagged TLR4. Cell lysates were subjected to Co-IP using anti-HA or anti-FLAG magnetic beads. Subsequently, Western blot analysis successfully detected FLAG-tagged TLR4 in HA-tagged TGR5 immunoprecipitates and vice versa, confirming a direct interaction. In contrast, no signals were detected in the corresponding single-transfection groups, confirming the specificity of the interaction.

These new findings from a controlled overexpression system provide direct biochemical evidence of the interaction, complementing our initial data from native proteins. We believe that combining these approaches validates the protein-protein interaction and strengthens the mechanistic foundation of our study. We are grateful for the reviewer's insightful comment, which have undoubtedly improved our work.

Response Fig. 5 *In vitro* Co-immunoprecipitation of TGR5 and TLR4 protein in HEK293T cells.

Comment 6: *In Fig.2J, 8 samples were limited to show the correlation between BSH and L. spp. The experimental number should be expanded.*

Response 6: Thank you very much for your valuable feedback. To address the concern regarding the limited sample size in the original correlation analysis between BSH activity and *Lactobacillus spp.* abundance (originally $n = 8$), we have expanded the analysis to include 17 samples.

The updated results reveals a strong and highly significant positive correlation between BSH activity and *Lactobacillus spp.* abundance ($r = 0.861$, $p < 0.001$), thereby strengthening the statistical validity of our initial finding. The revised scatter plot is provided here as Response Fig. 6 for your review and has been included in the manuscript as revised Fig. 2J (lines 142 – 143). This strong correlation further supports our conclusion and is consistent with the findings reported by Song et al. (2019) that *Lactobacillus* serves as a key source of BSH activity⁸.

We are grateful for this comment, which has improved the statistical power and quality of our study.

Response Fig. 6 Correlation analysis of *bsh* expression and the abundance of *Lactobacillus*.

Comment 7: *Discussion is necessary for the reason for the reduction of L. bacteria in the intestine of ICP patients.*

Response 7: Thank you for your insightful comment. As suggested, we have expanded the discussion in our manuscript (lines 295 – 309) to explore the reasons for the reduction in *Lactobacillus* abundance in ICP patients. The revised discussion primarily includes the following core points:

We propose that elevated systemic bile acid levels in ICP are the primary driving factor behind changes in the gut microbiota (particularly the reduction in *Lactobacillus*). Consistent with this view, Wu et al. (2024) demonstrated in a cholestatic mouse model that the abundance of *Lactobacillus* declined with increasing cholestatic severity⁹. The susceptibility of *Lactobacillus* to bile acids can be attributed to its Gram-positive nature¹⁰, a characteristic attributable to its simpler cell membrane structure compared to Gram-negative bacteria^{11,12}. Furthermore, *Lactobacillus* is not only a Gram-positive bacterium but also a major producer of bile acid hydrolase (BSH) activity in the gut, particularly the highly active BSH-T3 subtype, which is predominantly found in *Lactobacillus*⁸. Our results demonstrate a positive correlation between *Lactobacillus* abundance and BSH activity in the gut (manuscript Fig. 2I). This finding strongly suggests that the selective pressure of elevated bile acids on *Lactobacillus* leads to a reduction in BSH activity. This causal relationship is further strengthened by our fecal microbiota transplantation experiment, which shows a decrease in *Lactobacillus* and BSH activity, as well as elevated serum TBA, in recipient mice (manuscript Fig. 3). Collectively, these findings establish a self-reinforcing vicious cycle: elevated bile acids inhibit the growth of BSH-active bacteria, leading to abnormal bile acid metabolism, which further exacerbates bile acid accumulation.

We believe that the above supplementary discussion provides a clear and in-depth explanation for the key phenomenon of “reduced *Lactobacillus* in the gut of ICP patients”, fully addressing your concerns. Your feedback has greatly improved the quality of our manuscript, and we thank you for it.

References:

1. Lin, Q. et al. Intrahepatic cholestasis of pregnancy increases inflammatory susceptibility in neonatal

- offspring by modulating gut microbiota. *Front. Immunol.* **13**, (2022).
2. Hao, J. et al. Keratinocyte FABP5-VCP complex mediates recruitment of neutrophils in psoriasis. *Cell Rep.* **42**, 113449 (2023).
 3. He, Z. et al. The chlamydia psittaci inclusion membrane protein 0556 inhibits human neutrophils apoptosis through PI3k/AKT and NF- κ b signaling pathways. *Front. Immunol.* **12**, 694573 (2021).
 4. De Filippo, K., Henderson, R. B., Laschinger, M. & Hogg, N. Neutrophil chemokines KC and macrophage-inflammatory protein-2 are newly synthesized by tissue macrophages using distinct TLR signaling pathways. *J. Immunol.* **180**, 4308-4315 (2008).
 5. Ward, E. J. et al. Placental inflammation leads to abnormal embryonic heart development. *Circulation* **147**, 956-972 (2023).
 6. Luo, M. et al. Diagnostic and prognostic value of blood inflammation and biochemical indicators for intrahepatic cholestasis of pregnancy in chinese pregnant women. *Sci. Rep.* **12**, 20833 (2022).
 7. Eroğlu, H. et al. Increased serum delta neutrophil index levels are associated with intrahepatic cholestasis of pregnancy. *J. Obstet. Gynaecol. Res.* **47**, 4189-4195 (2021).
 8. Song, Z. et al. Taxonomic profiling and populational patterns of bacterial bile salt hydrolase (BSH) genes based on worldwide human gut microbiome. *Microbiome* **7**, (2019).
 9. Wu, L. et al. Lactobacillus acidophilus ameliorates cholestatic liver injury through inhibiting bile acid synthesis and promoting bile acid excretion. *Gut Microbes* **16**, 2390176 (2024).
 10. Horáčková, Š., Plocková, M. & Demnerová, K. Importance of microbial defence systems to bile salts and mechanisms of serum cholesterol reduction. *Biotechnol. Adv.* **36**, 682-690 (2018).
 11. Lambert, P. A. Cellular impermeability and uptake of biocides and antibiotics in gram-positive bacteria and mycobacteria. *J. Appl. Microbiol.* **92 Suppl**, 46S-54S (2002).
 12. Nikaido, H. Prevention of drug access to bacterial targets: permeability barriers and active efflux. *Science* **264**, 382-388 (1994).

Reviewer #2

Dear reviewer:

We sincerely thank you for your comprehensive evaluation and valuable feedback on the manuscript. The comments have been immensely helpful in strengthening our study. We have carefully considered each point raised and have performed extensive revisions to the manuscript to address them. We believe the manuscript has been substantially improved as a result of this process and hope that our responses and the revised manuscript now meet with your approval. Our point-by-point responses are provided below.

Comment 1: *There are awkward phrasing and incorrect word use as well as grammatical and typographical errors throughout the manuscript. Editing for grammar and syntax is needed.*

Response 1: Thank you very much for taking the time to review our manuscript and providing valuable suggestions for revision. In this revision, we have paid particular attention to optimizing the fluency of the language and the accuracy of the academic expression to ensure that the content is presented clearly, professionally, and in accordance with international academic publishing standards. To ensure that the revisions meet the highest standards, we have hired a professional academic English editing agency to comprehensively polish the language of the manuscript (Response Fig. 1). Should any language issues remain in the manuscript, such as English grammar errors, we kindly request that you inform us. We will endeavor to revise them to the best of our ability.

Response Fig. 1 Manuscript editing certificate.

Comment 2: *L77: conclude the Introduction with a statement of how the planned work is expected to advance the field.*

Response 2: We thank the reviewer for this suggestion. We have revised the final paragraph of the Introduction to conclude with a clear statement of our work's expected contribution to the field (**manuscript lines 53 – 58**).

This has been revised to the following in the manuscript:

“By investigating the relationship among the gut microbiome, bile acid metabolism, and ICP-associated placental inflammation, we aimed to identify candidate probiotics and explore their mechanisms in mitigating placental inflammation through TGR5 regulation. Our study provides mechanistic insights into ICP pathology and supports the development of microbiome-based therapies to improve fetal outcomes.”

This concluding statement now emphasizes that our research is expected to advance the field by providing new pathophysiological insights and exploring a novel therapeutic strategy, thereby significantly strengthening the introduction.

Comment 3: *Line 193, use a consistent bacterial name for LPG-9, is it Lactobacillus paragasseri or Lactobacillus paragallium?*

Response 3: We apologize for the error in the manuscript. We have performed a full check of the entire manuscript and have uniformly corrected all instances to “*Lactobacillus paragasseri* LPG-9”. This revision ensures consistency throughout the manuscript.

Comment 4: *L215-226: Figure panels of J-L are missing.*

Response 4: We sincerely apologize for this oversight. It was a significant omission. In the revised manuscript, we have completed the figure. The new Figure panels 4J, 4K, and 4L are now included (**manuscript Page 10**) and are accompanied by a detailed legend.

Comment 5: *Discussion: The discussion generally reiterates the results without much additional insight into interpretation or context for the study. Needs significant improvement.*

Response 5: We thank the reviewer for this critical comment. We have comprehensively rewritten the Discussion section to provide deeper mechanistic interpretation and a broader context for our findings, moving beyond a mere restatement of the results.

The main revisions are as follows.

(1) We not only described the changes in bile acids and microbiota in ICP but also delved into the underlying mechanisms (such as the selective toxicity of bile acids toward Gram-positive bacteria and the critical role of BSH activity), supported by extensive literature citations (**manuscript lines 290 – 309**).

(2) We integrated the interactions between gut microbiota dysbiosis, bile acid

metabolism disorders, and placental inflammation, proposing a “self-reinforcing vicious cycle” model, which may offer a new perspective for understanding the pathophysiology of ICP (**manuscript lines 310 – 329**).

(3) We have thoroughly discussed how the probiotics LPG-9 exerts its effects through the FXR-BSEP hepatic pathway and the TGR5 placental pathway, highlighting its specificity and application potential as an ICP intervention strategy (**manuscript lines 330 – 347**).

We believe that the revised Discussion has successfully provided an in-depth interpretation of the research results, explored the relevant mechanisms, and clearly articulated the progress made in our study. We appreciate the reviewer's insight, which has greatly strengthened the paper.

Comment 6: *L248-251: The reported association between LPG-9 supplementation and increased hepatic bile acid secretion is a significant finding that needs further discussion. Proposing a potential mechanistic pathway would strengthen the interpretation of this result.*

Response 6: We sincerely thank the reviewer for this insightful comment. We fully agree that elucidating the mechanistic pathway is crucial. As suggested, we have now expanded the discussion to propose a detailed mechanism by which LPG-9 promotes hepatic bile acid secretion, as described in the revised Discussion section (**manuscript lines 351 – 365**).

Briefly, we hypothesize that supplementation with LPG-9 increases BSH activity, leading to elevated levels of secondary bile acids and a reduction in the FXR antagonist T- β -MCA. The reduction of T- β -MCA alleviates inhibition on the FXR, resulting in the upregulation of the bile acid exporter Bile Salt Export Pump (BSEP). Enhanced BSEP activity accelerates the secretion of bile acids from the liver into bile ducts, thereby reducing the systemic bile acid load. Concurrently, the increase in secondary bile acids activates the TGR5 signaling pathway, which contributes to the mitigation of placental inflammation.

We believe this proposed pathway significantly strengthens the interpretation of our results and provides a theoretical foundation for the therapeutic application of targeted probiotics in ICP.

Comment 7: *L268: correct “signaling”*

Response 7: Thanks for your helpful suggestion. We have thoroughly reviewed the entire text and uniformly revised all instances of “signalling” to “signaling”.

Comment 8: *Line 318: Use full name for *Parabacteroides distasonis**

Response 8: Thank you for your valuable suggestions. In **line 312** of the revised version, “*P. distasonis*” has been changed to its full name “*Parabacteroides distasonis*”.

Comment 9: Line 342: Use full name for *Bacteroides fragilis*

Response 9: Thank you for your valuable suggestions. In line 348 of the revised version, “*B. fragilis*” has been changed to its full name “*Bacteroides fragilis*”.

Comment 10: Lines 378–380: Update the genus names for *Lactobacillus plantarum* and *Lactobacillus rhamnosus* to their current genus classifications, *Lactiplantibacillus plantarum* and *Lacticaseibacillus rhamnosus*, respectively. Please ensure these changes are applied consistently throughout the manuscript if the outdated names appear elsewhere.

Response 10: Thank you very much for pointing out this important taxonomic nomenclature update issue. We have conducted a comprehensive review and update of the genus names of *Lactobacillus* mentioned in the manuscript in accordance with the latest taxonomic nomenclature guidelines. The old genus names *Lactobacillus plantarum* and *Lactobacillus rhamnosus* have been updated to the latest official scientific names *Lactiplantibacillus plantarum* and *Lacticaseibacillus rhamnosus*. Additionally, we have thoroughly reviewed the entire manuscript (including the main text, figure captions, methods, and reference list) to ensure that all relevant strain names have been updated and are consistent throughout the document.

Comment 11: Data Analysis: Were the number of human placenta samples and number of animals per group were chosen based on power analysis? There is no info on the power analysis and no indication what outcome the study was powered or what is the primary outcome of the current study. The parameters (primary outcomes, and secondary outcomes if appropriate) used to estimate study power must be included.

Response 11: We sincerely thank the reviewer for raising this critical point regarding statistical power and sample size justification, which we acknowledge is essential for rigorous experimental design.

(1) Animal Studies:

Primary Outcome Selection:

The primary outcomes for the *in vivo* study were maternal TBA levels, survival rates and weight of offspring. These were selected based on their direct clinical relevance to ICP. Elevated maternal serum TBA concentration is a well-established, independent risk factor for adverse fetal outcomes, including fetal demise, preterm delivery, and low birth weight. Key mechanistic parameters in the placenta, including TGR5, TLR4, and NF- κ B signaling pathways and inflammatory cytokines, were pre-specified as secondary endpoints to explain the phenotypic observations.

Sample Size Justification:

The sample size for animal experiments was determined a priori using a power analysis. Based on effect sizes reported in similar published studies¹ ($f = 1.68$) corroborated by our pilot data ($d = 3.11$), we performed sample size calculations using G*Power 3.1 software. This calculation assessed statistical power for the prespecified primary

outcome measure. With $\alpha = 0.05$ and power $(1-\beta) = 0.80$, the analysis indicated that a sample size of $n = 3$ per group would be sufficient. To enhance robustness and account for potential variability, we used $n = 6$ per group. This size is also consistent with the results calculated using the resource equation approach under the 3R principle ($n = E/K + 1$)², and aligns with those commonly employed in similar studies in this field³⁻⁵. This information is stated in the Methods section (manuscript lines 400 – 402).

Based on the experimental results from this study, we further conducted a post-hoc power analysis. The analysis indicated that a sample size of $n = 6$ per group provides over 80% power for our primary outcomes, validating the sufficient sensitivity of our study (Response table. 1).

Response table. 1 Post-hoc power analysis conducted using G*Power

Experimental Groups	Number of groups	Parameters	Effect size f	Power (1- β)
Animal: Con vs 0.5%CA vs 1.0%CA	3	Serum TBA	2.21	1.00
Animal: Con vs 0.5%CA vs 1.0%CA	3	Survival rates of offspring	2.84	1.00
Animal: Con vs 0.5%CA vs 1.0%CA	3	Weight of offspring	0.89	0.87
Animal: Con vs. ICP vs. ICP+LPG-9	3	Serum TBA	2.21	1.00
Animal: Con vs. ICP vs. ICP+LPG-9	3	Survival rates of offspring	1.78	1.00
Animal: Con vs. ICP vs. ICP+LPG-9	3	Weight of offspring	0.92	0.89

(2) Human Placenta Samples:

We acknowledge the limited sample size in the human placenta study ($n = 5$ per group) and thank the reviewer for highlighting this. This limitation primarily stems from the relatively low incidence of ICP (1.5%–2.9%) and the stringent practical constraints associated with obtaining human tissue samples. Such a scale is not uncommon in exploratory research based on human tissue⁶. To enhance the reliability of our findings, all key observations were validated through multiple technical replicates using Western blotting and immunohistochemistry. Crucially, these results were replicated in our animal models with sufficient statistical power (manuscript Fig. 1G), providing strong corroborating evidence for our conclusions. Furthermore, Zhang et al. (2016) and Keitel et al. (2013) observed a significant downregulation of TGR5 mRNA and protein expression in the placentas of ICP patients and pregnant rats with cholestasis^{7,8}. A clinical study also found that the expression levels of TLR4 and NF- κ B in the placental tissue of pregnant women with ICP were positively correlated with the severity of ICP and adverse pregnancy outcomes⁹.

We have explicitly noted the sample size as a limitation in the Discussion section (manuscript lines 354 – 355) and are actively participating in a multi-center collaboration to expand the human cohort in future studies.

Comment 12: *qPCR analysis: What were the RNA integrity quality numbers for qPCR analyses? This helps to see if the integrity was high enough to get reliable qPCR results.*

Response 12: Thank you for your valuable suggestions. Throughout this study, we implemented a rigorous multi-step quality control process to ensure RNA quality and the reliability of subsequent qPCR data. All RNA samples were validated for purity via ultra-micro spectrophotometer, with A260/A280 ratios ranging between 1.9 and 2.1, and A260/A230 ratios consistently exceeding 2.0. The stability of the reference gene (GAPDH) was validated, with no significant differences in CT values across groups, indicating consistent reverse transcription efficiency. All qPCR amplifications exhibited a single peak in melting curve analysis, confirming amplification specificity and the absence of contaminants.

Following the reviewers' suggestions, we determined the 28S/18S ratio of RNA samples via agarose gel electrophoresis to assess RNA integrity. The samples showed signs of partial degradation due to prolonged storage, yet their 28S/18S ratios remained between 1.0 and 1.5 (Response Fig. 2). This indicates sufficient RNA integrity to yield reliable qPCR results.

Response Fig. 2 Agarose gel electrophoresis for assessing RNA integrity.

Comment 13: *Conclusion is a restatement of results without any integration or advancement in knowledge.*

Response 13: Thank you for your valuable feedback. Accordingly, we have completely rewritten the Conclusion to emphasize the integration of our findings and their advancement of knowledge in the field (manuscript lines 360 – 368).

The new conclusion emphasizes three key advancements. First, it defines the TGR5-TLR4-NF- κ B axis as a critical pathway in ICP pathogenesis. Second, it positions the probiotic LPG-9 as a novel dual-target therapeutic. Third, it establishes a new framework for microbiome-based intervention to improve pregnancy outcomes.

The revised text reads as follows:

“Our results revealed that gut microbiota dysbiosis, characterized by a reduction in BSH-active *Lactobacillus*, disrupts bile acid homeostasis, triggering placental

inflammation via the TGR5-TLR4-NF- κ B axis. Additionally, we demonstrated that intervention with BSH-enriched LPG-9 represents a novel dual-target strategy for ICP, which simultaneously activated hepatic BSEP to promote bile acid excretion and inhibited placental inflammation through TGR5. These findings not only reveal a critical mechanism of host-microbiome interactions in the pathogenesis of ICP, but also provide a transformative framework for developing novel microbiome-based therapies to improve pregnancy outcomes.”

We believe this revision effectively highlights the significance and transformative value of our work.

References

1. You, S. et al. Dysregulation of bile acids increases the risk for preterm birth in pregnant women. *Nat. Commun.* **11**, 2111 (2020).
2. Arifin, W. N. & Zahiruddin, W. M. Sample size calculation in animal studies using resource equation approach. *Malays. J. Med. Sci.* **24**, 101-105 (2017).
3. Barbosa, S. J. D. A. et al. The beneficial effects of lacticaseibacillus casei on the small intestine and colon of swiss mice against the deleterious effects of 5-fluorouracil. *Front. Immunol.* **13**, 954885 (2022).
4. Sandhu, M. et al. Friedelin attenuates neuronal dysfunction and memory impairment by inhibition of the activated JNK/NF- κ B signalling pathway in scopolamine-induced mice model of neurodegeneration. *Molecules* **27**, 4513 (2022).
5. Miljković, R. et al. Ameliorative effect of banana lectin in TNBS-induced colitis in c57BL/6 mice relies on the promotion of antioxidative mechanisms in the colon. *Biomolecules* **15**, 476 (2025).
6. Park, C. et al. Placental hypoxia-induced ferroptosis drives vascular damage in preeclampsia. *Circ. Res.* **136**, 361-378 (2025).
7. Zhang, Y. et al. Bile acids evoke placental inflammation by activating gpbar1/NF- κ B pathway in intrahepatic cholestasis of pregnancy. *J. Mol. Cell Biol.* **8**, 530-541 (2016).
8. Keitel, V. et al. Effect of maternal cholestasis on TGR5 expression in human and rat placenta at term. *Placenta* **34**, 810-816 (2013).
9. Xiao-Lin, L. Expression of NF- κ B, IL-17 and TLR4 in placenta of pregnant women with intrahepatic cholestasis of pregnancy and its relationship with pregnancy outcome. *Maternal and Child Health Care of China* **33**, 5092-5096 (2018).

Reviewer #1

Dear reviewer:

We sincerely thank you for your thorough review and constructive comments on our manuscript. We have carefully considered each point raised and have performed revisions to the manuscript to address them. We believe the manuscript has been substantially improved as a result and hope that our responses and the revised manuscript are now to your satisfaction. Our point-by-point responses are provided below.

Major comment 1: *The authors present that fecal matter from ICP patients can induce the condition (Fig. 3), while Lactobacillus paragallinarum LPG-9, potentially sourced from a healthy individual, appears to alleviate it (Fig. 4). These findings seem to present a significant conflict with the central hypothesis. Could the authors please clarify this apparent discrepancy or further discuss the scope and limitations of these results?*

Major response 1: Thank you for raising this important point, which provides us with an opportunity to clarify the logical progression between our key findings. The results presented in Fig. 3 and Fig. 4 are not contradictory. Instead, they collectively achieve two critical and complementary objectives of this study: establishing the pathogenic role of dysbiotic microbiota and validating a targeted therapeutic strategy.

(1)The FMT experiment confirms the pathogenic sufficiency of gut microbiota dysbiosis in ICP.

FMT establishes the causal relationship between gut microbiota dysbiosis and ICP placental inflammation. As shown in Fig. 3, transplanting gut microbiota from ICP patients into recipient mice reduces intestinal bile salt hydrolase (BSH) activity and induces key features of the disease. This transplantation system established a complete yet imbalanced gut ecosystem, where the increase of pathogenic bacteria was interlinked with the reduction of beneficial bacteria (As shown in Fig. R1, the abundance of *Lactobacillaceae* decreased while opportunistic pathogens such as *Desulfovibrionaceae* increased). These findings indicate that gut microbiota imbalance is sufficient to drive the progression of ICP.

Fig. R1 Pathogen proliferation accompanied by beneficial microbiota depletion following FMT.

(A) Bar diagram of the fecal microbial composition (Corresponding to Fig. 3C in the manuscript);

(B) Comparison of differentially abundant microbes.

Our mechanistic insight points to a critical loss of BSH activity in ICP. We propose that the pathogenicity of this gut microbiota dysbiosis may be driven by a

self-reinforcing cycle (Fig. R2, lines 292–334 in the Discussion). Elevated bile acid levels exert selective pressure on gram-positive gut bacteria, particularly BSH-enriched strains, resulting in a significant reduction in their BSH activity^[1-3] (a to c), lines 292–311 in the Discussion). Reduced BSH activity impairs the production of secondary bile acids (e.g., DCA and LCA) (c to d), thereby diminishing the endogenous activation of the bile acid receptor TGR5 and decreasing bile acid excretion^[4, 5] (d to a, lines 312–321 in the Discussion). This leads to further systemic bile acid accumulation, creating a self-perpetuating cycle that aggravates both gut microbiota dysbiosis and cholestasis.

Fig. R2 Schematic Diagram of a Self-Reinforcing Vicious Cycle Model

(2) Probiotic LPG-9 exerts its therapeutic effect by restoring deficient BSH activity.

The probiotic intervention serves as a therapeutic strategy aimed at addressing critical functional deficits observed in gut microbiota dysbiosis. We found that ICP-related dysbiosis is characterized by a significant reduction in BSH activity. As shown in Fig. R3, LPG-9 intervention significantly enhanced intestinal BSH activity and increased the relative abundance of *Lactobacillus*, indicating its successful colonization and functional benefit. Through its BSH activity, LPG-9 remodeled the bile acid profile, thereby alleviating placental inflammation via activating TGR5, and enhancing hepatic bile acid excretion via activating BSEP (Fig.6, lines 338–356 in the Discussion). This approach of utilizing probiotics to correct a specific dysfunction of the gut microbiota has been widely applied. For example, targeted supplementation with a specific strain capable of converting succinate into propionate has been shown to restore resistance against *Salmonella* colonization.^[6]

Fig. R3 LPG-9 intervention enhanced intestinal BSH activity, accompanied by an increased abundance of *Lactobacillus*.

(A) Principal coordinate analysis diagram of gut microbiome (Corresponding to Fig. 5A in the manuscript); (B) Bar diagram of the fecal microbial composition (Corresponding to Fig. 5B in the manuscript); (C) *bsh* expression in feces (Corresponding to Fig. 5D in the manuscript).

Logical integration of findings:

Together, our findings present a logical continuum. Fig.3 establishes gut microbiota dysbiosis as a sufficient cause of ICP (b to d), while Fig.4 demonstrates that a targeted intervention (LPG-9) correcting a specific functional deficit (BSH activity) can effectively treat it (interrupting c to d). These results are not contradictory but rather complementary and sequential. The former verifies the causal logic of the disease, while the latter identifies an effective therapeutic target. This approach aligns with that of Li et al., who used FMT to demonstrate that gut microbes can accelerate the progression of metabolic diseases, and subsequently achieved therapeutic effects by restoring function with specific bacteria^[7].

Scope and clinical translation:

We acknowledge that gut microbiota dysbiosis may influence ICP through multiple mechanisms. Our study specifically delineates a pathway centered on BSH activity. When the critical defect in the gut microbiota of ICP patients is reduced BSH activity in gut microorganisms, restoring BSH activity via probiotics supplementation constitutes an effective therapeutic strategy. Future studies should validate LPG-9 in clinical cohorts and investigate its interactions within the human microbiome.

In summary, the two key findings of our research are logically consistent. We have integrated this clarifying discussion and the aforementioned mechanistic model into the revised manuscript, and thank you for prompting this valuable refinement of our manuscript.

Major comment 2: *The authors report that TGR5 expression is decreased in ICP and that its agonist reduces inflammatory signaling. This raises a mechanistic question: how can an agonist be effective if the target receptor is downregulated? The authors*

should clarify the proposed mechanism. For instance, is the residual TGR5 expression sufficient for agonist-mediated signaling, or does the agonist work through an alternative pathway? This point should be addressed to strengthen the study's conclusions.

Major response 2: We appreciate your insightful question regarding the underlying mechanism. We propose that the agonist remains effective because the residual TGR5 expression in ICP is functionally competent and can be sufficiently activated. This conclusion is supported by (1) the inherent signal amplification of GPCR, (2) the unique signaling properties of TGR5, and (3) the high potency and selectivity of the agonist CCDC.

(1) Signal amplification compensates for reduced receptor quantity.

TGR5 is a Gs-protein-coupled receptor (GPCR). As with all GPCRs, TGR5 signaling benefits from intrinsic signal amplification, whereby a single activated receptor can initiate a cascade that generates abundant second messengers^[8, 9]. This fundamental principle supports that activating even a fraction of the total receptor pool can trigger a significant biological response. In ICP, while TGR5 expression is downregulated, it is not completely eliminated, as evidenced at both the protein and mRNA levels (Fig. R4). Therefore, the residual TGR5 receptors retain the inherent potential to generate a significant intracellular signal upon efficient activation.

Fig. R4 Downregulation of TGR5 in ICP, with residual expression at both protein and mRNA levels.

(A)Western blotting of placental TGR5 in human; (B)Western blotting of placental TGR5 in mice (Corresponding to manuscript Fig. 1G).

(2) Unique signaling properties maintain residual receptor efficacy.

Beyond universal amplification, TGR5 possesses unique properties that maintain signaling efficacy. Unlike many GPCRs, TGR5 exhibits resistance to rapid β -arrestin-mediated desensitization and shows delayed endocytosis, thereby maintaining prolonged signaling upon activation. Furthermore, TGR5 forms

preassembled heterodimers with EGFR within plasma membrane lipid rafts, a complex that enhances signal transduction efficiency^[10]. These characteristics suggest that residual TGR5 receptors can remain functionally responsive under pathological conditions. This is supported by our *in vitro* studies. In LPS-induced cellular models with downregulated TGR5 expression, agonist treatment not only restored TGR5 expression but also significantly reduced phospho-NF- κ B p65 levels (Fig. R5A). This demonstrates the retained capacity of residual TGR5 to transduce potent anti-inflammatory signals.

Fig. R5 CCDC activated TGR5 to exert anti-inflammatory effects *in vivo* and *in vitro* (A) Western blotting of TLR4 and NF- κ B in J774.2 cells treated with a TGR5 receptor agonist (Corresponding to manuscript Fig. 1J); (B) Western blotting of placental TGR5, TLR4 and NF- κ B (Corresponding to manuscript Fig. S2D).

(3) Highly selective and potent agonists ensure targeted activation.

The pharmacological efficacy of this approach is ensured by the agonist CCDC. The compound is a potent TGR5 agonist. Cellular assays demonstrate that CCDC strongly activates TGR5 ($pEC_{50} = 6.8$ and 7.5 in U2-OS cells and melanocytes, respectively) and translates into significant therapeutic efficacy, such as inhibiting inflammation in human primary monocytes ($pIC_{50} = 6.8$)^[11]. Furthermore, CCDC has been shown to effectively initiate signaling and produce therapeutic effects in other disease models characterized by decreased TGR5 expression^[12, 13]. Similarly, we directly confirmed CCDC activated the TGR5 pathway in the ICP animal model (Fig. R5B). This collective evidence validates the highly effective activation of the TGR5 pathway by CCDC in our ICP model.

CCDC is also a highly selective TGR5 agonist. Pharmacological profiling shows negligible activity at over 100 other targets, including the key bile acid receptor FXR^[11]. This high selectivity is further enhanced by the target tissue environment, where other bile acid receptors (FXR, PXR, and CAR) exhibit extremely low expression levels in both normal and cholestatic placentas^[14]. And our qPCR data confirmed that FXR, PXR, and CAR expression in the placenta was not significantly altered by CCDC intervention (Fig. R6), making them unlikely mediators of CCDC-induced responses in the target tissue. This result has been included in the manuscript as revised Fig. S2E. This collective evidence strongly supports TGR5 as

the principal target for CCDC intervention.

Fig. R6 mRNA expression levels of FXR, PXR, and CAR were not altered by CCDC intervention.

In summary, residual TGR5 receptors exhibit reduced quantity but maintain a robust quality of signaling, due to their inherent GPCR amplification effect combined with the unique signaling properties of TGR5. The highly specific agonist CCDC potently activates this remaining TGR5, thereby inducing anti-inflammatory responses. We have supplemented the discussion of the aforementioned mechanism in the manuscript's discussion section to strengthen the research conclusions (lines 281–291 in the Discussion).

Minor comment 1: *Since the clinical data from the five patients form the foundation of this study, it is crucial to present the key staining and Western blot results for all individual cases. Showing the complete dataset would strengthen the manuscript by demonstrating the reproducibility and consistency of the findings across the patient cohort.*

Minor response 1: We thank you for this constructive suggestion and fully agree. We have supplemented the revised manuscript with the key staining and Western blot results for all five ICP patients and five healthy controls. These results are presented in Fig. R7 for your review and has been included in the manuscript as revised Fig. S4 and Fig. S1B. The supplementary data visually demonstrate that the findings of this study exhibit high consistency within the patient cohort, such as downregulated TGR5 expression and activation of the TLR4-NF- κ B pathway. These additions significantly enhance the persuasiveness of the data and the robustness of the study conclusions.

Fig. R7 The key staining and Western blot results in all five ICP patients and five healthy controls. (A) Immunohistochemical analysis of TGR5, TLR4 and phospho-NF-κB p65 in the human placental tissue, 200× (Corresponding to manuscript Fig. S4); (B) Increased neutrophil infiltration in the placental tissues of patients with ICP (200×, MPO; Corresponding to manuscript Fig. S4); (C) Western blotting of

pro-inflammatory cytokines in the human placental tissue (Corresponding to manuscript Fig. S1B).

Minor comment 2: *For improved clarity and precision, we suggest that the authors revise the statement on line 131 to explicitly state that there was a decrease in *Lactobacillus* in the ICP group.*

Minor response 2: Thank you for the helpful feedback. We have revised the statement at **line 129** as suggested. The sentence has been amended to read: “**At the genus level, the abundance of *Lactobacillus* was significantly decreased in the ICP group compared to the control group.**” This revision makes the description more direct and precise, avoiding potential ambiguity.

Minor comment 3: *In Figure 2H, the label for one of the microbes appears to be overlapped, which makes it difficult to read. Please correct the label to ensure all text is clearly legible.*

Minor response 3: Thank you for pointing out the presentation issue in the figure. We have relocated the overlapping microbial labels and adjusted the font size to ensure all labels are fully legible. The figure is presented in Fig. R8 for your review and has been included in the manuscript as revised **Fig. 2H**.

Fig. R8 LEfSe Analysis of Gut Microbiota (Corresponding to manuscript Fig. 2H)

References:

- [1] Horáčková ` , Plocková M, Demnerová K. Importance of microbial defence systems to bile salts and mechanisms of serum cholesterol reduction[J]. Biotechnology Advances,2018,36(3):682-690.
- [2] Nikaido H. Prevention of drug access to bacterial targets: permeability barriers and active efflux[J].

Science,1994,264(5157):382-388.

[3] Song Z, Cai Y, Lao X, et al. Taxonomic profiling and populational patterns of bacterial bile salt hydrolase (BSH) genes based on worldwide human gut microbiome[J]. Microbiome,2019,7(1).

[4] Chen Q, Li Q, Cao M, et al. Hierarchy-Assembled Dual Probiotics System Ameliorates Cholestatic Drug-Induced Liver Injury via Gut-Liver Axis Modulation[J]. Adv Sci (Weinh),2022,9(17):e2200986.

[5] Zhao Q, Dai M Y, Huang R Y, et al. Parabacteroides distasonis ameliorates hepatic fibrosis potentially via modulating intestinal bile acid metabolism and hepatocyte pyroptosis in male mice[J]. Nat Commun,2023,14(1):1829.

[6] Wang Z, Kang S, Wu Z, et al. *Muribaculum intestinale* restricts *Salmonella* Typhimurium colonization by converting succinate to propionate[J]. The ISME Journal,2025,19(1):wraf069.

[7] Li L, Li T, Liang X, et al. A decrease in Flavonifractor plautii and its product, phytosphingosine, predisposes individuals with phlegm-dampness constitution to metabolic disorders[J]. Cell discovery,2025,11(1):25.

[8] Weis W I, Kobilka B K. The Molecular Basis of G Protein-Coupled Receptor Activation[J]. Annual review of biochemistry,2018,87:897-919.

[9] Berridge M J. Cell Signalling Pathways[M]. 6. 2014.csb0001002.

[10] Jensen D D, Godfrey C B, Niklas C, et al. The Bile Acid Receptor TGR5 Does Not Interact with β -Arrestins or Traffic to Endosomes but Transmits Sustained Signals from Plasma Membrane Rafts[J]. Journal of Biological Chemistry,2013,288(32):22942-22960.

[11] Evans K A, Budzik B W, Ross S A, et al. Discovery of 3-Aryl-4-isoxazolecarboxamides as TGR5 Receptor Agonists[J]. Journal of Medicinal Chemistry,2009,52(24):7962-7965.

[12] Caldwell A, Grundy L, Harrington A M, et al. TGR5 agonists induce peripheral and central hypersensitivity to bladder distension[J]. Scientific Reports,2022,12(1):9920.

[13] Bruce K, Zhang S, Garrido A N, et al. Pharmacological and physiological activation of TGR5 in the NTS lowers food intake by enhancing leptin-STAT3 signaling[J]. Nature Communications,2025,16(1):4990.

[14] Geenes V L, Dixon P H, Chambers J, et al. Characterisation of the nuclear receptors FXR, PXR and CAR in normal and cholestatic placenta[J]. Placenta,2011,32(7):535-537.

Reviewer #2

Dear reviewer:

We sincerely thank you for the positive feedback and for this helpful suggestion to enhance the methodological rigor of our manuscript.

A description of the RNA quality assessment has been added to the “RNA extraction and quantitative real-time PCR” subsection of the Methods section (**lines 479–484**), and have included the corresponding agarose gel electrophoresis results as revised **Fig. S5**.

We thank you once again for this suggestion, which has improved the completeness of our methodological reporting.